# HopaDIFF: Holistic-Partial Aware Fourier Conditioned Diffusion for Referring Human Action Segmentation in Multi-Person Scenarios

**Kunyu Peng**[1,2]    **Junchao Huang**[3]    **Xiangsheng Huang**[4,*]   **Di Wen**[1]    **Junwei Zheng**[1]
**Yufan Chen**[1]    **Kailun Yang**[5]    **Jiamin Wu**[6]    **Chongqing Hao**[7]    **Rainer Stiefelhagen**[1]
[1]Karlsruhe Institute of Technology    [2]INSAIT, Sofia University "St. Kliment Ohridski"
[3]Beijing Institute of Technology    [4]Institute of Automation, Chinese Academy of Sciences
[5]Hunan University    [6]Shanghai AI Lab    [7]HEBUST

## Abstract

Action segmentation is a core challenge in high-level video understanding, aiming to partition untrimmed videos into segments and assign each a label from a predefined action set. Existing methods primarily address single-person activities with fixed action sequences, overlooking multi-person scenarios. In this work, we pioneer *textual reference-guided* human action segmentation in multi-person settings, where a textual description specifies the target person for segmentation. We introduce the first dataset for Referring Human Action Segmentation, *i.e.*, **RHAS133**, built from 133 movies and annotated with 137 fine-grained actions with $33h$ video data, together with textual descriptions for this new task. Benchmarking existing action segmentation methods on **RHAS133** using VLM-based feature extractors reveals limited performance and poor aggregation of visual cues for the target person. To address this, we propose a *holistic-partial aware Fourier-conditioned diffusion* framework, *i.e.*, **HopaDIFF**, leveraging a novel cross-input gate attentional xLSTM to enhance holistic-partial long-range reasoning and a novel Fourier condition to introduce more fine-grained control to improve the action segmentation generation. **HopaDIFF** achieves state-of-the-art results on **RHAS133** in diverse evaluation settings. The dataset and code are available at https://github.com/KPeng9510/HopaDIFF.

## 1 Introduction

Human action recognition [6, 42, 38, 56, 45, 20] is a core problem with applications in robotics [44], surveillance [53], and healthcare [43]. While recent methods have shown strong performance, they typically rely on pre-trimmed video clips, which demand extensive manual pre-processing and limit applicability in real-time or continuous scenarios. Human action segmentation [14, 31, 57, 60], by contrast, operates on untrimmed video streams and aims to identify both action classes and their temporal boundaries. This setting is more challenging but aligns more closely with real-world use cases, where continuous and online interpretation of human behavior is essential.

However, existing human action segmentation approaches [40, 34, 2, 18] are predominantly designed for single-person scenarios, where the performed actions follow a predefined sequence dictated by a specific task protocol, *e.g.*, assembling a tool [47] based on fixed instructions or preparing a salad [50] according to a set menu. Such settings lack the complexity and variability present in real-world daily life scenarios, where multiple individuals may appear in the same untrimmed video, and the action sequences are not pre-defined but exhibit significant randomness and spontaneity. This limitation

---

*Correspondence: xiangsheng.huang@ia.ac.cn

39th Conference on Neural Information Processing Systems (NeurIPS 2025).

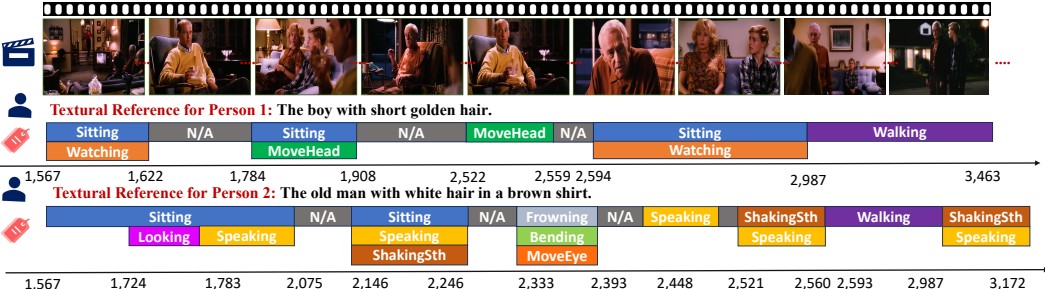

Figure 1: An illustration of the referring human action segmentation task is shown. The example is taken from the **RHAS133** dataset, with annotations provided for two referred individuals. The numbers beneath each annotation indicate the frame indices marking the start and end of each action. **N/A** denotes that the referred person is not present within the corresponding frame interval.

hinders the generalizability of current human action segmentation methods to more dynamic and unconstrained environments.

In this work, we introduce a novel task, termed *Referring Human Action Segmentation* (RHAS), which, to the best of our knowledge, is the first to address action segmentation in multi-person untrimmed videos guided by textual references. We contribute the **RHAS133** dataset, which comprises 133 untrimmed videos featuring 542 individuals, where each person of interest is annotated with a textual reference. This work is inspired by prior works on referring video and image understanding [42]. These textual descriptions serve as guidance for the model to localize and segment the actions of the referred individuals, with fine-grained, multi-label action annotations provided at the frame level following the AVA dataset format [21], as shown in Fig. 1.

We further adapt several well-established human action segmentation methods, *i.e.*, FACT [40], ActDiff [34], ASQuery [18], and LTContent [2], as well as one referring human action recognition method, *i.e.*, RefAtomNet [42], to the proposed RHAS setting and evaluate their performance on our newly constructed dataset, which serves as the first RHAS benchmark. However, the experimental results reveal that these baseline methods achieve limited performance under the RHAS framework. This can be attributed to the fact that existing human action segmentation approaches are primarily designed for single-person scenarios without textual guidance and are not equipped to handle unstructured action sequences without a fixed protocol in complex, multi-person movie environments. Meanwhile, RefAtomNet [45], originally developed for referring action recognition in trimmed videos, lacks the capacity for temporal modeling and long video understanding required by the RHAS task.

To improve referring human action segmentation, we introduce **HopaDIFF**, a novel framework that employs a holistic- and partial-aware Fourier conditioned diffusion strategy guided by untrimmed video and textual references. It consists of two branches: a holistic branch that captures global context from the full video, and a partial branch that leverages GroundingDINO [37] to detect the referred person, generating a cropped video stream for fine-grained representation using a Vision-Language Model (VLM), *e.g.*, BLIP-2 [29]. To facilitate effective long video reasoning and holistic-partial aware information exchange, we introduce a novel module, *i.e.*, HP-xLSTM, which fuses holistic and partial features using cross-input gate attentional xLSTM. Furthermore, to exert finer control over the generative process of the diffusion model for action segmentation, we incorporate an additional conditioning mechanism from the frequency domain based on the embeddings delivered by the HP-xLSTM. This is achieved by applying a Discrete Fourier Transform (DFT) along the temporal axis to extract frequency-based cues, which are combined with original feature conditions to guide the denoising process and are used to preserve temporal coherence and fine-grained details on motion dynamics by guiding the model to maintain consistency across both low and high-frequency components. The final segmentation prediction is obtained by averaging the denoised outputs from both branches, ensuring a robust and contextually informed result. Extensive experiments demonstrate that **HopaDIFF** achieves state-of-the-art performance on the proposed RHAS benchmark, significantly outperforming existing baselines in diverse evaluation settings.

To summarize, this paper makes three key contributions: (1) we introduce, for the first time, the task of Referring Human Action Segmentation (RHAS), paving the way for action segmentation in

complex, multi-person scenarios guided by natural language reference; (2) we present **RHAS133**, the first dataset for this task, and establish a comprehensive benchmark using diverse evaluation settings and adapted baseline methods; (3) we propose **HopaDIFF**, a diffusion-based framework tailored for RHAS, featuring the novel HP-xLSTM module for long-term temporal modeling through cross-input gate attention between holistic and partial branches together with additional Fourier condition. Our method achieves state-of-the-art performance on the **RHAS133** dataset.

## 2 Related Work

**Referring Scene Understanding.** Referring scene understanding, which leverages natural language to identify specific regions or objects within images or videos, has emerged as a valuable paradigm across various computer vision applications such as autonomous driving [58] and fine-grained action understanding [42]. The progress in this area has been greatly driven by the availability of high-quality, open-source datasets and benchmarks [5, 61, 52, 26, 48, 58, 8, 33], which provide standardized evaluation protocols and diverse referring scenarios. For instance, Li *et al.* [30] addressed referring image segmentation using a recurrent refinement network, whereas the CLEVR-Ref+ dataset introduced by Liu *et al.* [36] focused on visual reasoning with structured referring expressions. More recently, Wu *et al.* [58] proposed a benchmark for referring multi-object tracking, further broadening the scope of this research direction. Referring video segmentation [35, 59, 24, 65, 11, 17] is a related research direction, primarily targeting pixel-wise semantic segmentation of a specified object within trimmed videos. In contrast, our task focuses on per-frame action recognition to determine the temporal boundaries of different actions. Peng *et al.* [42] proposed the first benchmark to facilitate referring fine-grained human action recognition, yet their task is restricted to trimmed videos. In this work, we for the first time, conduct research on referring human action segmentation, which aims at achieving concrete action understanding on untrimmed videos. Since existing human action segmentation datasets [3, 27, 50] focused on single-person scenarios and consist of action sequences recorded according to a restricted protocol, which are suitable for the exploration of referring human action segmentation, we collect the first RHAS dataset, *i.e.*, **RHAS133**, from 133 different movies in multi-person scenarios with textual references. Our contributed dataset will serve as an important research foundation for the RHAS task.

**Video-based Human Action Segmentation.** Human action segmentation is a challenging video understanding task that involves dividing long, untrimmed sequences into frame-level action segments [10, 40, 34, 16, 63]. Unlike action recognition on trimmed clips with single labels, segmentation requires precise frame-wise predictions to model temporal boundaries. Common approaches leverage temporal convolutional networks [28, 22, 31, 19, 32, 41, 64], graph neural networks [25, 62], and transformers [60, 55, 12, 13, 51, 39] to model long-range dependencies. Multi-stage reasoning [60, 1, 31, 54, 14, 41, 57] further refines predictions iteratively. Recent methods such as FACT[40], sparse transformers[2], ASQuery [18], and diffusion models [34] have advanced the field. However, they are mostly designed for single-person settings with fixed action protocols, overlooking multi-person complexities. In this work, we tackle referring human action segmentation by adapting recognition models with vision-language features (*e.g.*, BLIP-2 [29], CLIP [46]) for textual guidance. We propose **HopaDIFF**, a diffusion-based model that fuses holistic and partial cues via cross-input gated attentional xLSTM and incorporates frequency-domain conditioning. Our method sets a new state of the art on the proposed **RHAS133** dataset.

## 3 Benchmark

**Dataset.** We present **RHAS133**, the first dataset specifically curated for the task of referring human action segmentation. It comprises 133 movies featuring diverse, multi-person scenarios. Annotations follow the AVA protocol [21], with the action label set extended from 80 to 137 fine-grained classes to capture a broader range of human behaviors. Textual references and action annotations are manually labeled and cross-validated by 6 annotators with domain expertise to ensure high annotation quality. The dataset spans 33 hours of video and includes 542 annotated individuals. Notably, the textual references describe target individuals without disclosing their actions.

Table 1 compares **RHAS133** with existing human action segmentation datasets. Prior datasets, such as SALADS50 [50], Assembly101 [47], EgoProceL [3], Breakfast [27], and GTEA [15], are typically limited to single-person settings, lack natural language references, and rely on predefined

Table 1: A statistical comparison with existing human action segmentation datasets, where **TextRef.** denotes the availability of textual reference annotations, **FreeProc.** indicates whether the dataset is collected under a free-form procedure rather than a fixed action sequence, and **NumClasses** and **NumPersons** represent the total number of action classes and the number of participants.

| Datasets | MultiPerson | TextRef. | MultiLabel | FreeProc. | View | Duration | NumClasses | NumPersons |
|---|---|---|---|---|---|---|---|---|
| SALADS50 [50] | ✗ | ✗ | ✗ | ✗ | Ego-centric | 4h | 50 | 25 |
| Assembly101 [47] | ✗ | ✗ | ✗ | ✗ | Ego-centric | 513h | 101 | 53 |
| EgoProceL [3] | ✗ | ✗ | ✗ | ✗ | Ego-centric | 62h | 16 | 130 |
| Breakfast [27] | ✗ | ✗ | ✗ | ✗ | Third-person | 77h | 10 | 52 |
| GTEA [15] | ✗ | ✗ | ✗ | ✗ | Ego-centric | 20h | 20 | 4 |
| Ours | ✓ | ✓ | ✓ | ✓ | Third-person | 33h | 137 | 542 |

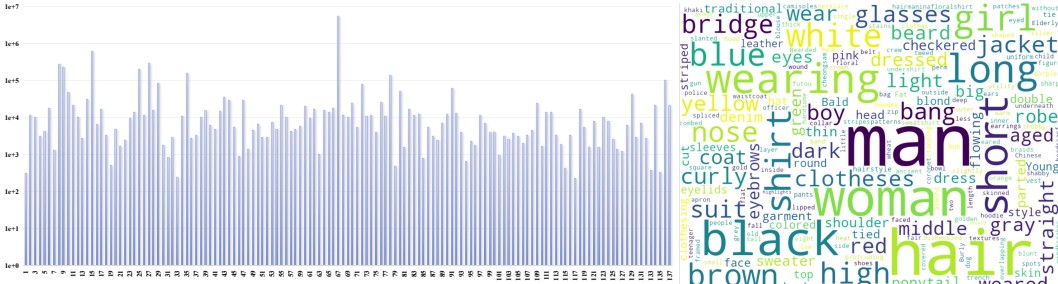

Figure 2: An illustration of the statistics of the dataset. The figure on the left-hand side shows the number of frames per action category, and the figure on the right is the word cloud generated based on the textual annotation in our **RHAS133** dataset. Zoom in for a better view.

procedural action sequences. The per-action class statistics and word cloud of the textual reference annotation are shown in Fig. 2. In contrast, **RHAS133** uniquely combines multi-person interactions, textual references, and fine-grained action annotations in unstructured, third-person video contexts. These properties make it a more comprehensive and challenging benchmark for advancing research in referring human action segmentation. Additional statistics are provided in the supplementary materials.

**Baselines.** To construct the first RHAS benchmark, we select baselines from both the human action segmentation and referring human action recognition domains to comprehensively evaluate performance under this novel setting. From the human action segmentation field, we include FACT [40], ActDiff [34], LTContent [2], and ASQuery [18], as they represent state-of-the-art methods employing diverse strategies such as dual-branch cross-attention (FACT), diffusion-based action segmentation model (ActDiff), sparse long-range transformer (LTContent), and query-based segmentation with boundary refinement (ASQuery). These models are chosen for their demonstrated ability to handle long, untrimmed videos and capture temporal dependencies, which are the key requirements for RHAS. In addition, we include RefAtomNet [42] from the referring human action recognition domain due to its effective use of vision-language models for textual grounding, which aligns well with the core challenge in RHAS. These selected baselines allow us to assess the capabilities and limitations of both traditional segmentation techniques and reference-guided approaches under the RHAS framework.

## 4 Methodology

### 4.1 Preliminaries.

Our proposed **HopaDIFF** is a diffusion-based action segmentation model, as shown in Fig. 3. Diffusion models [23, 49] aim to approximate the true data distribution $q$ through a parameterized model $p_\theta$. These models comprise two key components: the *forward process*, which progressively corrupts input data with Gaussian noise over $T$ time steps following a predefined variance schedule $\Gamma = \{\gamma_t \mid t \in [1, T]\}$, and the *reverse process*, which employs a neural network to iteratively reconstruct the data from noisy observations. During the forward process, the corrupted sample at time step $t$ is computed as $\mathbf{z}_t = \sqrt{\bar{\alpha}_t}\mathbf{z}_0 + \epsilon\sqrt{1 - \bar{\alpha}_t}$, where $\alpha_t = 1 - \gamma_t$ and $\bar{\alpha}_t$ denotes the

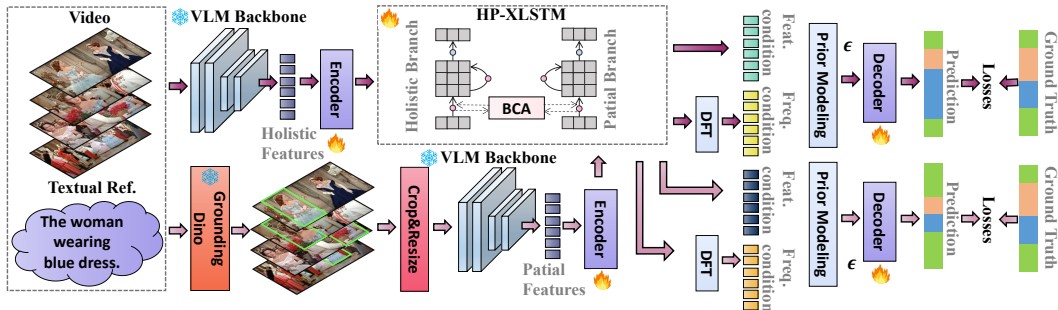

Figure 3: An overview of the proposed **HopaDIFF**, which integrates two complementary diffusion-based branches, *i.e.*, holistic and partial branches for action segmentation with target-referenced awareness. To improve controllability and segmentation precision, we introduce HP-xLSTM, a cross-input gated module designed for effective exchange between holistic and partial features, and propose a novel Fourier-based conditioning mechanism to inject frequency-domain control signals into the generative process. During training, the two branches are individually supervised using ground-truth action labels and temporal boundary annotations.

cumulative product of $\alpha_t$ over time. This formulation allows $\mathbf{z}_t$ to be directly sampled from the original data $\mathbf{z}_0$ in closed form, enabling a reparameterized expression of $\mathbf{z}$ suitable for optimization. The reverse process seeks to reverse this diffusion by gradually denoising the noisy sample at $T$ step back to a clean sample at $t = 0$ step. Each step of the reverse trajectory can be formulated as:

$$p_\theta(\mathbf{z}_{t-1}|\mathbf{z}_t) = \mathcal{N}(\mathbf{z}_{t-1}; \boldsymbol{\mu}, \sigma_t^2 \boldsymbol{I}), \tag{1}$$

where the mean $\boldsymbol{\mu}$ is predicted by a neural network, and $\sigma_t^2$ is determined by the variance schedule.

To model the denoising process, a neural network $\mathbf{M}_\theta$ is trained to estimate the original clean signal from its noise-corrupted counterpart. This network serves as the core of the reverse process by parameterizing the reconstruction procedure as defined in Eq. 2.

$$\mathbf{z}_{t-1} = \sqrt{\bar{\alpha}_{t-1}} \mathbf{M}_\theta(\mathbf{z}_t, t) + \sqrt{1 - \bar{\alpha}_{t-1} - \sigma_t^2} \frac{\mathbf{z}_t - \sqrt{\bar{\alpha}_t} \mathbf{M}_\theta(\mathbf{z}_t, t)}{\sqrt{1 - \bar{\alpha}_t}} + \sigma_t \epsilon. \tag{2}$$

Here, the timestep $t$ is sampled uniformly during training, and the loss encourages accurate reconstruction of $z_0$ from $z_t$. During inference, a sample $\mathbf{z}_0$ is generated by starting from Gaussian noise $\mathbf{z}_T \sim \mathcal{N}(\mathbf{0}, \boldsymbol{I})$ and iteratively applying the denoising step. Iterating Eq. 2 from $t = T$ to $t = 0$ yields a sample $\mathbf{z}_0$ from $p_\theta(\mathbf{z}_0)$. This formulation serves as the foundation for the existing diffusion-based action segmentation approach [34] and our **HopaDIFF**, enabling flexible conditioning and iterative refinement in the temporal domain.

## 4.2 HopaDIFF

**Diffusion-based Action Segmentation.** ActDiff [34] first adopted diffusion to achieve the action segmentation, which aims to learn how to denoise the action segmentation ground truth iteratively perturbed by Gaussian noise during training time and generate concrete action segmentation labels based on random noise and temporal feature conditions during inference. To capture the underlying distribution of human actions, the model is trained to reconstruct the original action sequence from its corrupted form. During each training iteration, a random diffusion step $t \in \{1, 2, ..., T\}$ is selected. Subsequently, noise is added to the original ground truth sequence $\mathbf{y}_0$ based on a cumulative noise schedule, resulting in a corrupted version $\mathbf{y}_t \in [0, 1]^{L \times C}$, which is defined as $\mathbf{y}_t = \sqrt{\bar{\alpha}_t} \mathbf{y}_0 + \epsilon \sqrt{1 - \bar{\alpha}_t}$ and $\epsilon$ denotes Gaussian noise. The denoising decoder $g_\psi$ is trained to handle sequences with varying noise levels, including fully random noise. In inference, it begins with a pure noise sample and progressively removes noise. The update at each step follows Eq. 2. This iterative process produces a sequence $[\hat{\mathbf{y}}_T, \hat{\mathbf{y}}_{T-1}, ..., \hat{\mathbf{y}}_0]$, with $\hat{\mathbf{y}}_0$ as the final prediction approximating the ground truth. Though ActDiff shows promising performance on the RHAS task, the lack of target-aware partial feature reasoning and the low granularity of control for the action segmentation generation limit its performance on the RHAS task.

To facilitate effective long-term information modeling and the exchange between holistic and partial cues related to the referred target, and to enhance the controllability of action segmentation generation conditioning, we propose a novel diffusion-based framework for referring human action segmentation, termed **HopaDIFF**. **HopaDIFF** is a novel diffusion-based framework for RHAS that integrates both holistic and partial video cues, guided by untrimmed video and corresponding textual references. It employs a dual-branch architecture, where the holistic branch captures global temporal context, and the partial branch focuses on fine-grained representations of the referred person using GroundingDINO [37] and VLM features from BLIP-2 [29]. To enhance temporal reasoning and controllability, **HopaDIFF** introduces a cross-input gate attentional module, *i.e.*, HP-xLSTM, for holistic-spatial feature exchange, and incorporates a frequency-domain conditioning mechanism using DFT to guide the diffusion-based action segmentation process.

**Holistic and Partial Conditioned Diffusion for RHAS.** HopaDIFF consists of two branches, *i.e.*, holistic and partial reasoning branches. For the holistic reasoning branch, the original video, *i.e.*, $\mathbf{v}_h$, is fed into a VLM-based cross-modal feature extractor (denoted as $\mathbf{F}_\phi$) together with textual reference, *i.e.*, $\mathbf{r}$, to achieve the preliminary cross modal feature extraction. For the partial reasoning branch, the video and textual reference are first fed into GroundingDino (G-Dino) [37] to achieve the detection of the person of interest, and then the cropped frames based on the bounding boxes prediction are grouped and resized to formulate a new video, fed into the same VLM feature extractor later. Two encoders ($\mathbf{E}_h$, $\mathbf{E}_p$) with the same structure are used for these two branches, where the parameters are not shared. Then the cross-modal feature conditions can be achieved by Eq. 3.

$$\mathbf{z}_h, \mathbf{z}_p = \mathbf{E}_h(\mathbf{F}_\phi(\mathbf{v}_h, \mathbf{r})), \mathbf{E}_p(\mathbf{F}_\phi(\text{G-Dino}(\mathbf{v}_h, \mathbf{r}), \mathbf{r})), \tag{3}$$

where $\mathbf{z}_h$ and $\mathbf{z}_p$ denote holistic cross-modal features and partial aware cross-modal conditions.

To achieve better long-range modeling and temporal feature aggregation, we propose holistic-partial aware xLSTM temporal reasoning, which can achieve cue exchange between both the holistic branch and the partial branch through bidirectional cross-input gate attention, abbreviated as HP-xLSTM. The enhanced features from those two branches can be calculated according to Eq. 4

$$\hat{\mathbf{z}}^h, \hat{\mathbf{z}}^p = \text{HP-xLSTM}(\mathbf{z}^h, \mathbf{z}^p). \tag{4}$$

Next, we will introduce the details regarding our proposed HP-xLSTM.

**Holistic-Partial Aware xLSTM Temporal Reasoning.** Extended LSTM (xLSTM) [4] introduces two major components, *i.e.*, exponential gating and enhanced memory structures. These give rise to two variants, *i.e.*, sLSTM, which uses scalar memory with mixing, and mLSTM, which employs matrix memory with a parallelizable outer-product update. While mLSTM drops recurrent memory mixing which can be used to enable parallelism, sLSTM supports multiple cells and heads with selective memory mixing across cells. The holistic partial feature exchange is conducted at the input gate of the mLSTM block, as Eq. 5.

$$
\begin{aligned}
\boldsymbol{C}_t^m &= f_t^m \boldsymbol{C}_{t-1}^m + \mathrm{i}_t^m \boldsymbol{v}_t^m \, \boldsymbol{k}_t^{m\top}, m \in \{h, p\} & & \text{cell states} \\
\boldsymbol{n}_t^m &= \mathrm{f}_t^m \boldsymbol{n}_{t-1}^m + \mathrm{i}_t^m \boldsymbol{k}_t^m & & \text{normalizer state} \\
\boldsymbol{h}_t^m &= \mathbf{o}_t^m \odot \tilde{\boldsymbol{h}}_t^m, \qquad \tilde{\boldsymbol{h}}_t^m = \boldsymbol{C}_t^m \boldsymbol{q}_t^m \, / \, \max\left\{ \left| \boldsymbol{n}_t^{m\top} \boldsymbol{q}_t^m \right|, 1 \right\} & & \text{hidden state} \\
\boldsymbol{q}_t^m &= \boldsymbol{W}_q^m \, \boldsymbol{z}_t^m v^m + \boldsymbol{b}_q^m & & \text{query input} \\
\boldsymbol{k}_t^m &= \frac{1}{\sqrt{d}} \boldsymbol{W}_k^m \, \boldsymbol{z}_t^m + \boldsymbol{b}_k^m, & & \text{key input} \\
\boldsymbol{v}_t^m &= \boldsymbol{W}_v^m \, \boldsymbol{z}_t^m + \boldsymbol{b}_v^m & & \text{value input} \\
\mathrm{i}_t^m &= \exp\left( \tilde{\mathrm{i}}_t^m \right), \qquad \tilde{\mathrm{i}}_t^h, \tilde{\mathrm{i}}_t^p = \mathbf{BCA}\left( \boldsymbol{w}_\mathrm{i}^{h\top} \, \boldsymbol{z}_t^h + b_\mathrm{i}^h, \, \boldsymbol{w}_\mathrm{i}^{p\top} \, \boldsymbol{z}_t^h + b_\mathrm{i}^h \right) & & \text{input gate} \\
\mathrm{f}_t^m &= \sigma\left( \tilde{\mathrm{f}}_t^m \right) \text{ OR } \exp\left( \tilde{\mathrm{f}}_t^m \right), \quad \tilde{\mathrm{f}}_t^m = \boldsymbol{w}_{\mathrm{f}^m}^{m\top} \, \boldsymbol{z}_t^m + b_\mathrm{f} & & \text{forget gate} \\
\mathbf{o}_t^m &= \sigma\left( \tilde{\mathbf{o}}_t^m \right), \qquad \tilde{\mathbf{o}}_t^m = \boldsymbol{W}_{\mathbf{o}^m}^m \, \boldsymbol{z}_t^m + \boldsymbol{b}_\mathbf{o}^m & & \text{output gate,}
\end{aligned}
\tag{5}
$$

where BCA indicates the bidirectional cross-attention conducted on the input gates of the holistic and partial feature reasoning branches.

**Discrete Fourier Transformation based Frequency Conditions.** Since action segmentation fundamentally relies on per-frame action recognition to produce temporally coherent segmentation

outputs, a high degree of control granularity is essential when employing diffusion models to generate segmentation results from Gaussian noise, particularly when using features as conditions. Existing approaches, *e.g.*, ActDiff [34], primarily leverage features in the temporal and spatial domains as conditioning inputs, while overlooking the potential advantages offered by the frequency domain. In this work, we explicitly incorporate frequency-domain information, derived from features aggregated via HP-xLSTM, to enhance the fine-grained controllability of the diffusion process and the capability of the model to deal with complex periodic temporal patterns. The frequency-based conditions for both the holistic and partial branches are computed according to Eq. 6.

$$\hat{\mathbf{z}}_f^h, \ \hat{\mathbf{z}}_f^p = \text{DFT}(\hat{\mathbf{z}}^h), \ \text{DFT}(\hat{\mathbf{z}}^p), \tag{6}$$

where DFT indicates the discrete Fourier transform. We adopt two decoders, *i.e.*, $\mathbf{D}_h$ and $\mathbf{D}_p$, with the same structure for both the holistic and partial branches. During training, the label interrupted by Gaussian noise, $\mathbf{y}_t$, is used as input to the decoder together with the spatial, temporal, and frequency embedding conditions as shown in the following equation.

$$\mathbf{s}^h, \mathbf{s}^p = \mathbf{D}_h(\mathbf{y}_t, \hat{\mathbf{z}}_f^h, \mathbf{z}_f^h), \mathbf{D}_p(\mathbf{y}_t, \hat{\mathbf{z}}_f^p, \mathbf{z}_f^p), \tag{7}$$

where the predicted action segmentation results from those two branches are merged together during the inference time. The outputs are supervised directly by the original action segmentation annotations using binary cross-entropy loss and temporal boundary loss, which is used to align the action boundaries in the denoised sequence and the ground truth sequence.

## 5 Experiments

### 5.1 Implementation Details

All the experiments are conducted on 1 NVIDIA A100 GPU and PyTorch 2.0.0. We adopt both BLIP-2 [29] and CLIP [46] as the feature extractors, for all the experiment tables shown in the main paper, we set frame length as $3,000$ for each sample to get the train/val/test sets, where cross-movie evaluation denotes that the videos used for the aforementioned three sets come from different movies, which measures both of the cross-subject generalizability and cross-scenario generalizability. On the other hand, the random partition denotes that the samples of the train/val/test sets are randomly selected. For the cross-movie evaluation settings, the train/val/test sets consist of $566$, $61$, and $163$ long-video samples, respectively. For the random evaluation settings, the train/val/test sets consist of $626$, $82$, and $82$ long-video samples, respectively. Each sample has a time length of 3.3 minutes. The input feature dimension of the encoder is $768$ for BLIP-2 [29] feature extractor and is $512$ for CLIP [46] feature extractor, where we use ASFormer (12 layers with 256 feature maps) as the feature encoder and ASFormer (8 layers with 128 feature maps) according to [60, 34]. All the models are trained using the Adam [9] optimizer with a batch size of 4. The learning rate to train our model is selected as $1e-4$. The binary cross-entropy loss and the boundary loss are equally weighted. All the baselines in this paper adopt holistic input apart from the experiments reported in Table 6.

In line with prior studies in the field of human action segmentation [34, 40, 2], we evaluate model performance using frame-wise accuracy (ACC), edit score (EDIT), and segmental F1 scores at overlap thresholds $\{10\%, 25\%, 50\%\}$ (F1@10, 25, 50). The *ACC* metric captures the accuracy of the predictions at the individual frame level, whereas the EDIT and F1 scores focus on the temporal consistency and quality of action segmentation between segments. Note that, our approach is a two-stage approach, where we rely on pre-extracted VLM features.

### 5.2 Benchmark Analysis

We first conduct training and evaluations on the random partition of the RHAS dataset and use BLIP-2 [29] as the feature extractor, where the experimental results are shown in Table 2. The referring action recognition approach, *i.e.*, RefAtomNet [42], shows the second-best ACC, *i.e.*, $39.48\%$, while its performances of F1@$\{10, 25, 50\}$ are limited, since it is not specifically designed for long video understanding and lacks long temporal reasoning capability. Most of the human action segmentation baselines delivers better F1@$\{10, 25, 50\}$, where LTContent and ActDiff achieve $> 60\%$ of F1@$\{10, 25, 50\}$, thanks to their promising long-term dependency modeling capability. Among all the human action segmentation baselines, ActDiff [34] shows best performance for the referring human action

Table 2: Experimental results with BLIP-2 [29] cross-modal feature extractor and under random partition setting on the RHAS dataset.

| Method | Val | | | | | Test | | | | |
|---|---|---|---|---|---|---|---|---|---|---|
| | ACC | EDIT | F1@10 | F1@25 | F1@50 | ACC | EDIT | F1@10 | F1@25 | F1@50 |
| FACT [40] | 26.30 | 0.27 | 55.86 | 54.06 | 50.38 | 26.08 | 0.27 | 52.91 | 50.77 | 47.06 |
| ActDiff [34] | 42.01 | 7.05 | 70.74 | 68.19 | 63.66 | 41.85 | 7.20 | 70.56 | 68.34 | 63.29 |
| ASQuery [18] | 31.37 | 0.10 | 35.21 | 33.90 | 31.94 | 32.30 | 0.12 | 35.59 | 33.58 | 29.51 |
| LTContent [2] | 35.72 | 0.30 | 68.70 | 66.40 | 62.25 | 34.23 | 0.31 | 64.70 | 63.09 | 58.50 |
| RefAtomNet [42] | 39.48 | 0.12 | 36.22 | 34.68 | 32.48 | 38.01 | 0.13 | 34.01 | 31.93 | 27.62 |
| Ours | **62.69** | **7.50** | **89.07** | **86.23** | **80.91** | **62.58** | **7.75** | **87.96** | **85.50** | **79.39** |

Table 3: Experimental results with BLIP-2 [29] cross-modal feature extractor and under cross-movie evaluation setting on the RHAS dataset.

| Method | Val | | | | | Test | | | | |
|---|---|---|---|---|---|---|---|---|---|---|
| | ACC | EDIT | F1@10 | F1@25 | F1@50 | ACC | EDIT | F1@10 | F1@25 | F1@50 |
| FACT [40] | 26.01 | 0.34 | 60.16 | 57.79 | 55.33 | 27.89 | 0.44 | 59.05 | 58.37 | 57.13 |
| ActDiff [34] | 4.46 | 5.96 | 38.86 | 38.36 | 37.51 | 2.36 | 15.09 | 22.44 | 22.28 | 21.80 |
| ASQuery [18] | 22.95 | 0.05 | 34.39 | 31.52 | 27.96 | 20.84 | 0.08 | 33.86 | 32.86 | 30.20 |
| LTContent [2] | 49.49 | 0.33 | 42.69 | 40.12 | 37.22 | 52.52 | 0.37 | 49.35 | 47.24 | 42.55 |
| RefAtomNet [42] | 35.00 | 0.10 | 49.58 | 46.60 | 41.58 | 38.73 | 0.13 | 41.16 | 39.17 | 35.44 |
| Ours | **54.41** | **6.31** | **89.14** | **86.56** | **83.35** | **59.63** | **19.37** | **90.91** | **90.33** | **89.26** |

Table 4: Experiments with CLIP [46] and under random partition on **RHAS133**.

| Method | Val | | | | | Test | | | | |
|---|---|---|---|---|---|---|---|---|---|---|
| | ACC | EDIT | F1@10 | F1@25 | F1@50 | ACC | EDIT | F1@10 | F1@25 | F1@50 |
| FACT [40] | 23.99 | 0.46 | 56.66 | 54.95 | 51.06 | 23.95 | 0.44 | 55.10 | 53.63 | 49.72 |
| ActDiff [34] | 13.31 | 7.28 | 29.88 | 28.84 | 25.92 | 13.21 | 7.38 | 29.19 | 28.19 | 25.68 |
| ASQuery [18] | 32.59 | 0.07 | 29.63 | 28.52 | 26.97 | 26.35 | 0.06 | 25.08 | 23.82 | 19.79 |
| LTContent [2] | 13.20 | 2.08 | 35.29 | 30.96 | 28.24 | 13.99 | 1.78 | 30.96 | 29.20 | 22.38 |
| RefAtomNet [42] | 27.27 | 0.10 | 29.38 | 28.25 | 25.92 | 27.80 | 0.11 | 29.27 | 27.86 | 24.41 |
| Ours | **46.99** | **7.68** | **77.00** | **74.39** | **69.57** | **46.62** | **7.44** | **74.74** | **72.28** | **73.16** |

Table 5: Ablations on **RHAS133** using BLIP-2 [29] and under partition setting on different movies.

| Method | Val | | | | | Test | | | | |
|---|---|---|---|---|---|---|---|---|---|---|
| | ACC | EDIT | F1@10 | F1@25 | F1@50 | ACC | EDIT | F1@10 | F1@25 | F1@50 |
| w/o all | 4.46 | 5.96 | 38.86 | 38.36 | 37.51 | 2.36 | 15.09 | 22.44 | 22.28 | 21.80 |
| w/o HCMGB | 20.78 | 6.34 | 45.81 | 43.93 | 41.95 | 23.44 | 19.93 | 50.60 | 50.14 | 49.42 |
| w/o HP-XLSTM | 47.75 | 6.30 | 82.81 | 80.61 | 77.63 | 52.01 | 19.68 | 85.06 | 84.54 | 83.50 |
| w/o BCAF | 44.24 | **6.35** | 79.76 | 77.38 | 74.50 | 49.56 | 19.87 | 82.98 | 82.45 | 81.38 |
| w/o DFT-cond | 50.84 | 6.33 | 85.88 | 83.47 | 80.52 | 55.49 | 19.92 | 87.95 | 87.42 | 86.38 |
| FACT [40] | 26.01 | 0.34 | 60.16 | 57.79 | 55.33 | 27.89 | 0.44 | 59.05 | 58.37 | 57.13 |
| FACT [40] w/ partial branch | 18.35 | 0.32 | 62.57 | 61.10 | 61.10 | 18.34 | 0.43 | 63.72 | 63.35 | 62.40 |
| ActDIFF [34] | 4.46 | 5.96 | 38.86 | 38.36 | 37.51 | 3.26 | 15.09 | 22.44 | 22.28 | 21.80 |
| ActDIFF [34] w/ DFT-cond | 10.46 | 6.29 | 26.61 | 24.80 | 23.40 | 11.92 | 19.54 | 30.01 | 29.64 | 28.92 |
| Cross-Attention Fusion [7] | 51.74 | **6.41** | 86.91 | 84.47 | 81.50 | 57.02 | **19.87** | 88.97 | 88.41 | 87.36 |
| Ours | 54.41 | 6.31 | **89.14** | **86.56** | **83.35** | 59.63 | 19.37 | **90.91** | **90.33** | **89.26** |

segmentation task, which delivers $42.01\%$ of ACC, $7.05\%$ of EDIT, $70.74\%$, $68.19\%$, abd $63.66\%$ of F1@{10, 25, 50} on the validation set and $41.85\%$ of ACC, $7.20\%$ of EDIT, $70.56\%$, $68.34\%$, and $63.29\%$ of F1@{10, 25, 50} on the test set.

Diffusion-based human action segmentation benefits from its diffusion-based framework, which iteratively refines action predictions from noise, enabling it to capture complex temporal dependencies and action priors, which benefits the cue reasoning of the referring human action segmentation task. However, we observe that limited performance is delivered by ActDiff [34] on the cross-movie evaluation setting, as shown in Table 3. One underlying reason is that its learned action priors,

*e.g.*, temporal position and relational dependencies, are subject- and scenario-specific and may not transfer effectively to movies with different scene structures, pacing, or narrative conventions. Among all the baselines used in the benchmark, LTContent [2] produces the best ACC as shown in Table 3, *i.e.*, 49.49%, while FACT [40] delivers best performances in terms of F1@{10, 25, 50}, *i.e.*, 60.16%, 57.79%, and 55.33% on the validation set and 59.05%, 58.37%, and 57.13% on the test set. Regarding the cross-movie evaluation, most of the benchmarked baselines suffer performance decays in terms of F1@{10, 25, 50}, indicating that cross-scenario and cross-subject setting is challenging for most of the human action segmentation baselines in the RHAS task, while the referring human action recognition baseline RefAtomNet [42] works better in cross-movie evaluation scenario, thanks to its cross-branch agent attention mechanism which could will address the information fusion between local and global cues. ActDiff delivers the best EDIT score on both the random partition and cross-movie partition due to its iterative denoising mechanism, which enhances temporal consistency and boundary precision across varying data distributions. Since our task focuses on fine-grained action annotations, recognizing precise action boundaries is inherently difficult, leading to lower EDIT scores.

Our proposed **HopaDIFF** outperforms existing diffusion-based action segmentation methods by incorporating holistic-partial feature fusion through HP-xLSTM and introducing frequency-domain conditioning, which together enhance target-aware reasoning and enable more precise and controllable action segmentation in complex multi-person scenarios. **HopaDIFF** delivers 62.69% of ACC, 7.50% of EDIT, 89.07%, 80.91%, and 69.57% of F1@{10, 25, 50} on the validation set and 62.58% of ACC, 7.75% of EDIT, 87.96%, 85.50%, and 79.39% of F1@{10, 25, 50} on the test set on random partition. We can also observe that **HopaDIFF** delivers similar F1@{10, 25, 50} on the cross-movie evaluation setting, which overcomes the shortcomings of the existing diffusion-based human action segmentation approach by using high control granularity provided by holistic-partial aware xLSTM-based temporal reasoning and the frequency condition.

We further evaluate the cross VLM backbone generalizability of all the leveraged baselines and our proposed **HopaDIFF** method in Table 4. We first observe that most of the leveraged approaches show better performances when they use the BLIP-2 [29] cross-modal feature extractor compared with the CLIP [46] cross-modal feature extractor, indicating that using a strong VLM backbone can benefit the RHAS task. Among all compared methods, our proposed **HopaDIFF** achieves the best performance across every metric on both validation and test sets. Specifically, **HopaDIFF** reaches 46.99% ACC, 7.68% EDIT, and 77.00%, 74.39%, 69.57% F1 scores on the validation set, and 46.62% ACC, 7.44% EDIT, and 74.74%, 72.28%, 73.16% F1 scores on the test set. These results surpass all baselines by large margins, particularly in temporal consistency and segmental precision. In contrast, the best baseline FACT only achieves 23.99% ACC and a maximum F1@50 of 49.72% on the test set, indicating the difficulty of using a weaker VLM backbone in our RHAS task.

Notably, **HopaDIFF** maintains stable performance across diverse evaluation settings, highlighting its generalization capability. This performance gain is attributed to its dual-branch design that fuses holistic and partial cues, and the integration of Fourier-based frequency conditioning for finer generative control. Overall, the strong results confirm that **HopaDIFF** effectively addresses the unique challenges of referring human action segmentation in complex, multi-person scenarios. We further provided the number of trainable parameters, inference times, and GFLOPS of baselines and our HopaDIFF in Table 7 for efficiency comparison.

### 5.3 Ablation Analysis

The ablated module variants include: (1) *w/o all*, a vanilla diffusion-based action segmentation model without any of the proposed components; (2) *w/o HCMGB*, which removes the GroundingDINO [37]-based partial reasoning branch; (3) *w/o HP-xLSTM*, which omits the holistic-partial aware xLSTM temporal aggregation module; (4) *w/o BCAF*, which excludes the bidirectional cross-attention fusion between the input gates of the two branches; and (5) *w/o DFT-cond*, which eliminates frequency-domain conditioning in both branches. The full **HopaDIFF** model achieves the highest performance, with 59.63% ACC, 19.37% EDIT, and 90.91%, 90.33%, 89.26% F1 scores at thresholds 10%, 25%, and 50% on the test set. Removing HP-xLSTM leads to a significant drop in F1@50 from 89.26% to 83.50%, demonstrating the importance of long-range holistic-partial temporal modeling. Eliminating DFT-cond lowers F1@50 to 86.38%, validating the effectiveness of frequency-based conditioning in improving the precision and controllability of action segmentation generation. The absence of BCAF

Table 6: Ablation experiments for partial branch, when we use BLIP-2 and cross-movie partition.

| Method | Val | | | | | Test | | | | |
|---|---|---|---|---|---|---|---|---|---|---|
| | ACC | EDIT | F1@10 | F1@25 | F1@50 | ACC | EDIT | F1@10 | F1@25 | F1@50 |
| FACT [40] | 16.73 | 0.39 | 66.07 | 65.73 | 64.69 | 15.82 | 0.30 | 57.30 | 55.75 | 53.72 |
| ActDIFF [34] | 3.84 | 3.51 | 51.38 | 50.53 | 47.87 | 2.04 | 7.67 | 44.13 | 43.70 | 42.42 |
| ASQuery [18] | 32.42 | 0.11 | 51.18 | 48.31 | 43.32 | 36.51 | 0.12 | 44.46 | 42.52 | 39.09 |
| LTContent [2] | 45.36 | 0.08 | 60.51 | 57.00 | 50.87 | 50.34 | 0.08 | 49.58 | 47.14 | 42.43 |
| RefAtomNet [42] | 27.73 | 0.13 | 46.90 | 43.52 | 38.82 | 30.45 | 0.15 | 38.01 | 36.30 | 32.72 |
| Ours w/ only PB | 44.48 | 6.28 | 79.40 | 77.34 | 74.42 | 48.45 | **19.70** | 81.99 | 81.49 | 80.48 |
| **Ours w/ HPB** | **54.41** | **6.31** | **89.14** | **86.56** | **83.35** | **59.63** | 19.37 | **90.11** | **90.33** | 89.26 |

also causes a performance decrease, with F1@50 dropping to $81.38\%$, highlighting the utility of cross-branch interaction through input gate attention. Notably, removing the GroundingDINO [37]-based partial reasoning branch (HCMGB) reduces F1@50 to $49.42\%$, showing that person-specific local cues are essential for accurate target-aware segmentation. Overall, the ablation results confirm that each proposed component, *i.e.*, GroundingDINO [37]-based partial cue reasoning, HP-xLSTM-based fusion, and frequency conditioning, plays a critical role in our model, and their synergy leads to the superior performance of **HopaDIFF** on the complex textual referring action segmentation task. The experiments also reveal that replacing the action branch with the partial branch in FACT ((6) *FACT [40] w/ partial branch*) improves F1@{10,25,50} by better capturing long-duration, movement-related actions, but decreases ACC due to the loss of broader contextual cues. Adding a frequency-domain condition to ActDIFF ((7) *ActDIFF [34] w/ DFT-cond*) enhances boundary detection (higher EDIT) but shows weaker generalization across movies without the HP-xLSTM module, leading to lower F1 on validation. Finally, while cross-attention fusion performs competitively, our HP-xLSTM achieves superior accuracy and F1 by enabling structured, fine-grained temporal modeling and selective bidirectional information exchange between holistic and partial branches. We further test cropping the person region detected by GroundingDino before feeding it into the baselines (Table 6). Compared to holistic inputs (Table 3), this results in lower ACC but partial gains in F1, showing the complementary roles of holistic and partial branches: the holistic branch captures background and interaction cues, while the partial branch focuses on the target individual to improve movement-related action recognition.

# 6 Conclusion

In this work, we introduced Referring Human Action Segmentation (RHAS), a new task aimed at segmenting actions of individuals in multi-person untrimmed videos using natural language references. To support this, we constructed the **RHAS133** dataset with fine-grained action labels and referring expressions. We proposed **HopaDIFF**, a dual-branch diffusion model that fuses global and localized cues via HP-xLSTM and DFT-based conditioning for improved temporal reasoning and control. Extensive experiments show that **HopaDIFF** outperforms existing baselines in both accuracy and temporal consistency. Our contributions pave the way for fine-grained, language-guided video understanding in complex settings.

# Acknowledgements

The project is funded by the Deutsche Forschungsgemeinschaft (DFG, German Research Foundation) – SFB 1574 – 471687386. This work was also partially supported by the SmartAge project sponsored by the Carl Zeiss Stiftung (P2019-01-003; 2021-2026). This work was performed on the HoreKa supercomputer funded by the Ministry of Science, Research and the Arts Baden-Württemberg and by the Federal Ministry of Education and Research. The authors also acknowledge support by the state of Baden-Württemberg through bwHPC and the German Research Foundation (DFG) through grant INST 35/1597-1 FUGG. This project is also supported in part by the National Natural Science Foundation of China under Grant No. 62473139, in part by the Hunan Provincial Research and Development Project (Grant No. 2025QK3019), and in part by the Open Research Project of the State Key Laboratory of Industrial Control Technology, China (Grant No. ICT2025B20). This research was partially funded by the Ministry of Education and Science of Bulgaria (support for INSAIT, part of the Bulgarian National Roadmap for Research Infrastructure).

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

# Appendix

## A  Society Impact and Limitations

### A.1  Society Impact

In this work, we introduce Referring Human Action Segmentation (RHAS), a novel task that enables fine-grained action understanding in complex, multi-person video scenes using natural language descriptions. We contribute the first dataset for this task, **RHAS133**, along with a diffusion-based referring human action segmentation solution, **HopaDIFF**, which integrates global and localized cues to achieve precise, target-aware action segmentation. These contributions lay important groundwork for the future of language-guided video understanding.

This research has the potential to significantly impact several real-world domains. In healthcare and eldercare, RHAS can enable systems to achieve action segmentation in shared spaces with multiple persons when a textual reference is given for a specific person, *e.g.*, "the elderly man in the red shirt". In human-robot interaction, robots could better understand and track human behaviors in dynamic group settings. Similarly, in surveillance or safety monitoring, RHAS may help systems localize and interpret behaviors based on spoken or written descriptions, offering more intuitive and flexible interaction with video analytics tools.

However, this technology also raises ethical considerations. First, the reliance on data from publicly available movies may embed social, cultural, or gender biases present in media, which could affect model fairness and generalizability. For instance, actions associated with underrepresented groups may be poorly modeled, leading to biased or inaccurate outputs. Second, RHAS systems might be misused for invasive surveillance or profiling if deployed without safeguards, raising privacy and civil liberty concerns.

To mitigate these risks, future work should focus on curating more diverse, representative datasets and embedding fairness-aware training objectives. Transparency, stakeholder oversight, and responsible deployment frameworks will be critical to ensuring that the benefits of RHAS technologies are equitably distributed and ethically aligned with societal values.

### A.1.1  Limitations

As stated in our main paper, our work is the first to investigate human action segmentation guided by textual references to enable action segmentation for a specific individual in multi-person scenarios. Due to the novelty of the task, experiments are currently limited to our constructed dataset, which constrains the evaluation of the proposed model's generalizability. Nevertheless, we made substantial efforts to ensure dataset diversity, collecting data from a wide range of movies over an annotation process that spanned more than 8 months. To further assess generalizability, we designed multiple evaluation protocols, *e.g.*, random partition and cross-movie partition, and conducted cross-backbone experiments by replacing BLIP-2 [29] with CLIP. In future work, we aim to expand the dataset by incorporating a larger and more diverse set of movies. Our method is a two-stage method that relies on pre-extracted VLM features, limiting the efficiency of inference. However, we also note that most existing methods (*e.g.*, FACT, ActDIFF, LTContext, ASQuery) also adopt a two-stage processing pipeline due to the high computational cost of long video processing. We acknowledge this point also as an open challenge in action segmentation field.

## B  Further Clarification of the BCA Mechanism

The BCA module enables information exchange between holistic and partial branches by using cross-input-gate attention to modulate input information for each stream jointly, supporting richer temporal reasoning. HP-xLSTM leverages cross-input gated attention (*i.e.*, BCA applied to input gates), enabling each branch to selectively incorporate temporal cues from the other based on feature-level gating. This design enhances temporal segmentation and boosts per-frame accuracy, particularly in challenging multi-person scenarios with fine-grained actions.

The attention weights and fused representations are computed as follows:

$$\alpha_t^{H \to P} = \mathrm{softmax}\left( \frac{(W_Q X_t^H)(W_K X_t^P)^\top}{\sqrt{d}} \right), \tag{8}$$

$$\tilde{X}_t^P = \alpha_t^{H \to P} \cdot (W_V X_t^H), \tag{9}$$

$$\alpha_t^{P \to H} = \mathrm{softmax}\left( \frac{(W_Q X_t^P)(W_K X_t^H)^\top}{\sqrt{d}} \right), \tag{10}$$

$$\tilde{X}_t^H = \alpha_t^{P \to H} \cdot (W_V X_t^P), \tag{11}$$

where $W_Q, W_K, W_V \in \mathbb{R}^{d \times d}$ are learnable projection matrices for query, key, and value. $\tilde{X}_t^P$ and $\tilde{X}_t^H$ are the cross-attended inputs of xLSTM input gates in the corresponding branches.

## C   More Implementation Details

In this section, we will provide more implementation details regarding our proposed HopaDIFF framework. For both encoders of the holistic branch and the partial branch, the number of feature maps is chosen as $64$, and the kernel size is chosen as $5$.

We adopt a normal dropout rate of $0.5$, a channel dropout rate of $0.5$, and a temporal dropout rate of $0.5$. For both the decoders of the holistic branch and the partial branch, we choose the dimension of time embedding as $512$, the number of feature maps as $24$, the kernel size as $5$, and the dropout rate as $0.1$. Regarding the hyperparameters of the diffusion process, we choose the timesteps as $1,000$

Table 7: Efficiency measurement when we leverage BLIP-2 feature extractor on RHAS133, cross-movie partition.

| Method | Trainable Param. | Inference Time | GFLOPS |
|---|---|---|---|
| FACT [40] | 10.29M | 1.98s | 16.37G |
| ACTDIFF [34] | 7.7M | 2.28s | 7.14G |
| ASQuery [18] | 47.28M | 1.24s | 57.22G |
| LTContent [2] | 6.12M | 2.70s | 20.02G |
| RefAtomNet [42] | 106M | 3.29s | 273.53G |
| HopaDIFF | 20M | 2.81s | 10.72G |

and the sampling time steps as $25$. The training epoch is set as $1,000$. The number of parameters, inference time, and GFLOPS of the baseline approaches and our approach are reported in Table 7.

## D   Ablation Study of the frame length

In this subsection, we provide further ablation regarding the frame length for the data partition, where we change the setting used in the main paper from $3,000$ to $2,000$, where the results are demonstrated in Table 8. We could observe that our proposed HopaDIFF outperforms the others with large margins under this frame length setting due to its superior capability to achieve highly controllable human action segmentation generation. This ablation further illustrates the superior generalization capability of **HopaDIFF** model.

Table 8: Experimental results with BLIPv2 cross-modal feature extractor and under cross-movie evaluation setting on the RHAS dataset, using frame length 2000.

| Method | Val | | | | | Test | | | | |
|---|---|---|---|---|---|---|---|---|---|---|
| | ACC | EDIT | F1@10 | F1@25 | F1@50 | ACC | EDIT | F1@10 | F1@25 | F1@50 |
| FACT [40] | 38.06 | 0.44 | 75.26 | 73.88 | 70.52 | 36.95 | 0.50 | 73.89 | 72.84 | 70.62 |
| ActDiff [34] | 4.59 | 25.48 | 38.96 | 38.59 | 37.66 | 3.00 | 54.96 | 27.41 | 27.08 | 26.14 |
| ASQuery [18] | 36.12 | 0.12 | 31.40 | 29.70 | 25.91 | 34.12 | 0.12 | 35.24 | 33.24 | 28.25 |
| LTContent [2] | 12.80 | 0.55 | 29.12 | 27.88 | 24.40 | 13.65 | 0.59 | 31.18 | 29.18 | 25.15 |
| RefAtomNet [42] | 29.47 | 0.12 | 32.80 | 31.00 | 27.14 | 33.62 | 0.15 | 46.29 | 44.13 | 39.32 |
| **Ours** | **59.60** | **35.00** | **94.33** | **92.62** | **88.85** | **62.63** | **90.78** | **94.71** | **93.57** | **91.19** |

## E   Ablation regarding the Dual Head Setting

Training two branches separately leads to better performance compared with the mentioned counterpart, as shown in Table 9. It presents an ablation study comparing the Merged baseline with

our dual-head design. On the validation set, our approach achieves higher accuracy (54.41% vs. 52.12%) and consistently outperforms across all F1 metrics, with improvements of +2.04%, +2.05%, and +1.86% at F1@10, F1@25, and F1@50, respectively. On the test set, our method similarly shows superior accuracy (59.63% vs. 56.50%) and notable gains in F1 scores across all thresholds. Although the merged setting achieves slightly higher EDIT (6.42 vs. 6.31 on validation, 19.89 vs. 19.37 on test), our dual head design provides a better balance, enhancing both per-frame accuracy and temporal overlap performance. These results highlight the advantage of explicitly modeling holistic and partial branches separately rather than merging them, as the dual head structure allows more fine-grained temporal reasoning and better recognition of actions across different contexts.

Table 9: Ablation study of the dual head setting.

| Method | Val | | | | | Test | | | | |
|--------|-----|------|-------|-------|-------|------|------|-------|-------|-------|
| | ACC | EDIT | F1@10 | F1@25 | F1@50 | ACC | EDIT | F1@10 | F1@25 | F1@50 |
| Merged | 52.12 | **6.42** | 87.10 | 84.51 | 81.49 | 56.50 | **19.89** | 88.28 | 87.69 | 86.64 |
| **Ours** | **54.41** | 6.31 | **89.14** | **86.56** | **83.35** | **59.63** | 19.37 | **90.11** | **90.33** | **89.26** |

## F  Ablation regarding the Diffusion Action Segmentation Mechanism

To evaluate the contribution of the proposed diffusion mechanism in our human action segmentation framework, we conduct an ablation study by removing the diffusion module and comparing the results against our full model. The quantitative results are reported in Table 10, covering both validation and test sets across multiple metrics.

Without the diffusion mechanism, the model demonstrates a clear degradation in performance across all metrics. On the validation set, the accuracy drops from 54.41% to 53.43%, while the EDIT score declines drastically from 6.31 to 0.09, indicating poor temporal alignment. Similarly, F1 scores decrease significantly, *e.g.*, from 89.14% to 67.35% at F1@10 and from 83.35% to 81.50% at F1@50. A similar trend is observed on the test set, where the absence of diffusion reduces the EDIT score from 19.37 to 0.10, and segmental F1 scores experience large drops, such as from 90.33% to 56.17% at F1@25.

These results highlight the importance of the diffusion mechanism for our model. The large improvement in the EDIT score shows that diffusion plays a critical role in refining temporal boundaries and improving sequence coherence. Furthermore, the consistent gains across all F1 metrics confirm that diffusion enhances the model's ability to capture fine-grained action transitions, ultimately leading to more robust and accurate action segmentation.

Table 10: Ablation study of the diffusion-based human action segmentation mechanism.

| Method | Val | | | | | Test | | | | |
|--------|-----|------|-------|-------|-------|------|------|-------|-------|-------|
| | ACC | EDIT | F1@10 | F1@25 | F1@50 | ACC | EDIT | F1@10 | F1@25 | F1@50 |
| w/o Diffusion | 53.43 | 0.09 | 67.35 | 63.68 | 81.50 | 57.47 | 0.10 | 58.56 | 56.17 | 53.72 |
| **Ours** | **54.41** | **6.31** | **89.14** | **86.56** | **83.35** | **59.63** | **19.37** | **90.11** | **90.33** | **89.26** |

## G  Ablation regarding the Efficacy of Different Branches

To further investigate the contribution of each component in our framework, we conduct an ablation study on the partial and holistic branches. The results are summarized in Table 11, where we compare the performance of our full model against two variants: one without the partial branch and one without the holistic branch.

When the partial branch is removed, the performance drops drastically across all metrics. On the validation set, the accuracy decreases from 54.41% to 20.78%, and F1 scores suffer severe degradation, *e.g.*, from 89.14% to 45.81% at F1@10 and from 83.35% to 41.95% at F1@50. A similar trend is observed on the test set, where accuracy falls from 59.63% to 23.44%, and F1@25

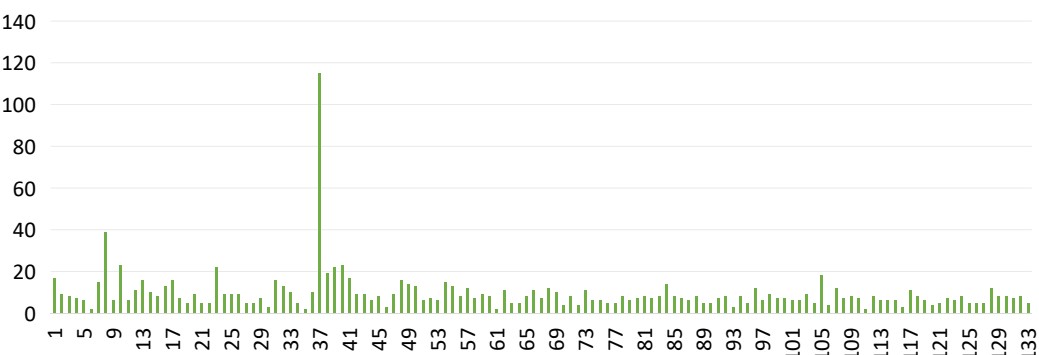

Figure 4: An overview of the statistics regarding the number of persons per movie in our **RHAS133** dataset. The horizontal axis denotes the video ID, and the vertical axis denotes the number of persons annotated in the corresponding video.

decreases sharply from $90.33\%$ to $50.14\%$. These results highlight the critical role of the partial branch in capturing fine-grained and localized action cues.

Removing the holistic branch also results in performance degradation, though the drop is less severe than removing the partial branch. On the validation set, accuracy decreases to $44.48\%$, while F1@10 and F1@25 decline to $79.40\%$ and $77.34\%$, respectively. On the test set, the accuracy reduces to $48.45\%$, with F1@25 dropping from $90.33\%$ to $81.49\%$. This indicates that the holistic branch contributes to capturing global temporal dependencies and overall action context, thereby improving segmentation coherence.

Overall, these ablations demonstrate that both the partial and holistic branches play complementary roles in our method. The partial branch is essential for modeling detailed action transitions, while the holistic branch ensures global consistency. Their combination enables our model to achieve state-of-the-art segmentation performance.

# H    Qualitative Results

We further provide a visualization of qualitative results of our approach and the FACT model based on one sample from the **RHAS133** dataset in Figure 5. We observe that **HopaDIFF** consistently produces action segmentation outputs that more closely align with the ground truth labels. Notably, our method demonstrates superior performance in handling periodic action transitions along the temporal axis. This improvement can be attributed to the high granularity of the condition offered by the Fourier embeddings, especially on periodic temporal patterns, and the enhanced temporal reasoning capabilities of our proposed holistic-partial-aware cross-input-gate xLSTM architecture for different visual focuses. We also identified a performance drop in classes with limited training samples. In such long-tail scenarios, frames corresponding to rare classes are often misclassified as more frequent classes. We acknowledge this limitation and consider tackling the long-tail distribution challenge in referring human action segmentation as a promising and important direction for future research.

Table 11: Ablation study of different branches in our method.

| Method | Val | | | | | Test | | | | |
|---|---|---|---|---|---|---|---|---|---|---|
| | ACC | EDIT | F1@10 | F1@25 | F1@50 | ACC | EDIT | F1@10 | F1@25 | F1@50 |
| w/o partial branch | 20.78 | **6.34** | 45.81 | 43.93 | 41.95 | 23.44 | **19.93** | 50.60 | 50.14 | 49.42 |
| w/o holistic branch | 44.48 | 6.28 | 79.40 | 77.34 | 74.42 | 48.45 | 19.70 | 81.99 | 81.49 | 80.48 |
| **Ours** | **54.41** | 6.31 | **89.14** | **86.56** | **83.35** | **59.63** | 19.37 | **90.11** | **90.33** | **89.26** |

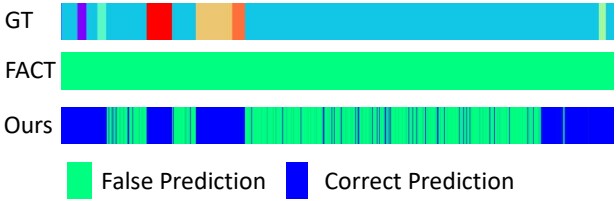

Figure 5: Qualitative results of our **HopaDIFF** and FACT baseline, where false predictions are marked as green and correct predictions are marked as blue. Each color shown in GT denotes a different set of combinations of atomic-level actions.

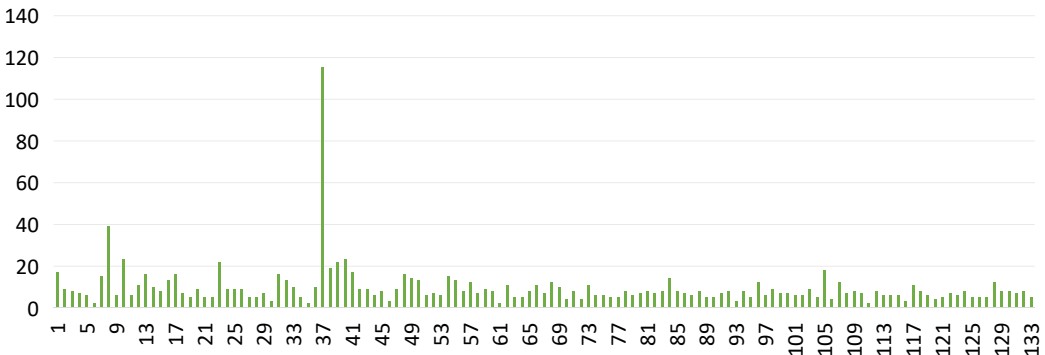

Figure 6: An overview of the statistics regarding the number of persons per movie in our **RHAS133** dataset. The horizontal axis denotes the video ID, and the vertical axis denotes the number of persons annotated in the corresponding video.

# I   More Details of the RHAS133 Dataset

In this section, we deliver more details of our contributed **RHAS133** dataset. First, we present the number of persons statistic for each movie in Fig. 6, which illustrates that the distribution of our dataset is diverse in terms of the number of persons per video.

