# OpenReview forum: "HopaDIFF: Holistic-Partial Aware Fourier Conditioned Diffusion for Referring Human Action Segmentation in Multi-Person Scenarios"
_NeurIPS.cc/2025/Conference — NeurIPS 2025 spotlight_

### Official Review · Reviewer_dRDv · 2025-06-19

**Clarity:** 2
**Significance:** 4
**Originality:** 2
**Rating:** 5
**Confidence:** 4

**Summary:**

This paper pioneers a novel task—Referring Human Action Segmentation (RHAS)—which extends traditional single-person temporal action segmentation to multi-person, textual reference-guided scenarios that are inherently more unconstrained and unstructured. The authors introduce RHAS133, the first dataset for this task, comprising 133 untrimmed movies with 542 annotated individuals. Each person is described with textual references and labeled with fine-grained action annotations. Furthermore, the authors propose HopaDIFF, a holistic-partial aware Fourier-conditioned diffusion framework that integrates dual-branch modeling, HP-xLSTM with bidirectional cross-attention (BCA), diffusion-based generation, and frequency-domain conditioning. The proposed approach significantly outperforms both conventional temporal action segmentation and referring action recognition baselines.

**Questions:**

1.The end of the Introduction section would benefit from a clear summary of the main contributions of this work.

2.Regarding the dataset, there is a significant class imbalance and long-tailed distribution: some action categories contain over 1e+6 frames, while others have fewer than 1e+3. Although such imbalance is common in many datasets, the authors should explain the extent of this imbalance in detail, discuss its impact on model performance, and describe any mitigation strategies adopted—or justify the choice not to address it. This is critical for ensuring transparency and the credibility of the results.

3.The Methodology section lacks sufficient architectural details for several modules. Specific concerns include:
A. The model appears to follow a two-stage pipeline—first extracting Holistic and Partial features, then performing subsequent modeling—but this is not clearly stated or described in the paper.
B. For the denoising diffusion model, the internal structure of the denoising function $M_{\theta}$ is not detailed. The paper does not explain how the model processes the noisy input, timestep, and the spatial, temporal, and frequency embedding conditions during denoising.
C. The Bidirectional Cross-Attention (BCA) module lacks a thorough explanation of its mechanism and role within the architecture.

4.Concerns of the Innovation:
A. While the use of existing components (e.g., VLM and Grounding DINO for feature extraction) is technically reasonable, key modules in the proposed framework—such as the diffusion-based segmentation and xLSTM—are borrowed directly from prior work. The paper allocates significant space and equations to these existing methods, while offering limited discussion of the truly novel aspects such as dual-branch interaction and frequency-conditioned denoising, which weakens the perceived originality.
B. The explanations of the novel components remain superficial. For instance: 1) The rationale for using frequency-based conditioning is not sufficiently elaborated; the current explanation ("primarily leverage features in the temporal and spatial domains...") does not justify why frequency information is beneficial for denoising. 2) The mechanism of BCA-based information exchange is not clearly explained.  3) The decision to keep the two branches separate (rather than fuse them) and train them independently is not justified, nor is this design explored through ablation studies.

5.Efficiency Issues: The paper lacks a discussion of efficiency in terms of parameter count, FLOPs, and training/inference time. The use of two VLM backbones, one Grounding DINO, two ASFormer encoders and decoders, and a diffusion model—each computationally heavy—raises concerns about the feasibility and efficiency of the two-stage pipeline.

6.Performance and Fairness Issues: In Table 5, the performance of the model without HCMGB is still lower than that of FACT, despite both relying solely on holistic features. This raises questions about the comparative effectiveness of other modules in HopaDIFF, and an explanation is needed.

7.Experimental Issues:
A.Why do the test set results exceed those on the validation set in most experiments?
B. The overall EDIT scores are low; a justification is needed.
C. Table 5 lacks an ablation study for removing the diffusion model entirely.
D. There are no component-level ablation studies examining the internal structure or parameters of each individual module.
E. There is no analysis of training each branch independently or of fusing the branches—such ablations are essential for validating the dual-branch design.

8.Errors and Inconsistencies:
Line 104: "state of the art" should be hyphenated as "state-of-the-art".
Lines 85–88: The sentence beginning with “since...” and ending with “which...” is overly long and should be split for clarity.
Line 283: The reported F1@50 score is inconsistent with the corresponding table.
Table 5 and Lines 310 & 315 refer to “w/o DCT-cond,” which suggests Discrete Cosine Transform, while the earlier sections describe frequency conditioning based on DFT (Discrete Fourier Transform); this inconsistency needs to be clarified.

This work presents a task and framework with acceptable levels of innovation and effort. However, there remain multiple critical concerns regarding methodology, clarity, originality, and experimental completeness. I assign a Borderline Accept and will adjust my score based on the authors’ responses and revisions.

**Ethical Concerns:**

["NO or VERY MINOR ethics concerns only"]

**Final Justification:**

Within the limited space, the author has provided thorough and detailed responses that sufficiently address the majority of my concerns. These revisions have significantly improved the manuscript's quality, clarity, and overall rigor. Therefore, I recommend accepting this paper.

**Limitations:**

The paper discusses dataset limitations and future directions in both the conclusion and supplementary materials, and provides a fairly comprehensive analysis of the societal impact, including both positive and negative aspects. However, it lacks a clear discussion of the limitations of the proposed framework itself.

**Paper Formatting Concerns:**

No Formatting Concerns

**Quality:**

3

**Strengths And Weaknesses:**

Quality: This work initiates the RHAS task, proposes the RHAS133 dataset, and develops the HopaDIFF framework. The technical methodology is sound, and experiments demonstrate state-of-the-art performance in this emerging area. However, the paper lacks sufficient ablation studies and does not provide an in-depth analysis of each innovative component.
Clarity: The motivation and overall structure are clearly presented. However, critical architectural details are missing, limiting reproducibility. For instance, the interaction mechanism between the two branches within HP-XLSTM via BCA, the internal structure and information flow of the denoising model at each diffusion step, and how spatial, temporal, and frequency embedding conditions guide the model are not sufficiently explained.
Significance: This work pioneers the RHAS task, expanding action segmentation into a more complex, textual-reference-guided multi-person scenario, and introduces a corresponding dataset and framework. The contribution is of high significance to the field.
Originality: The paper introduces a novel research task and a new dataset. While the proposed framework includes several innovative modules, many components (e.g., diffusion model, xLSTM) are adapted or combined from existing works. Thus, while the task-level originality is high, the architectural novelty is relatively limited.

---

> ### Author Rebuttal · Authors · 2025-07-30
>
> Dear Reviewer dRDv,
>
> Thank you for the review effort and valuable comments. We provide point-to-point responses as follows,
>
> > **Q1**-Introduction summary.
>
> We will add the following summary into the end of the introduction section.
>
> **To summarize, this paper makes three key contributions:
> (1) we introduce, for the first time, the task of Referring Human Action Segmentation (RHAS), paving the way for action segmentation in complex, multi-person scenarios guided by natural language; (2) we present RHAS133, the first dataset for this task, and establish a comprehensive benchmark using diverse evaluation settings and adapted baseline methods; (3) we propose HopaDIFF, a diffusion-based framework tailored for RHAS, featuring the novel HP-xLSTM module for long-term temporal modeling through cross-input gate attention between holistic and partial branches together with additional fourier condition. Our method achieves state-of-the-art performance on the RHAS133 benchmark.**
>
> > **Q2**-Long-tailed distribution problem.
>
> Thank you for your comment. **Our dataset is not collected according to a fixed protocol and is based on in-the-wild data, which results in the unbalanced and long-tailed issue. In-the-wild data typically exhibits a long-tailed distribution because real-world activities naturally vary in frequency, common actions (e.g., walking, talking) occur far more often than rare ones (e.g., fainting, waving a flag) [1].** This imbalance poses a challenge for the referring human action segmentation pipelines, which tend to favor frequent classes while underperforming on rare but often critical ones. Despite this, in-the-wild data is crucial for developing robust and generalizable models, as it captures the diverse, unstructured, and noisy nature of real-world environments [2]. **In order to mitigate the negative effects we use class weight with inverse relationship to the frequency of each class to weight the loss function.** Training and evaluating on in-the-wild data improves a model's practical applicability in unconstrained settings for human action segmentation, such as surveillance, healthcare, or human-robot interaction. We will modify the manuscript accordingly to deliver the above discussion.
>
> References:
>
> [1] Yang, Lu, et al. "A survey on long-tailed visual recognition." IJCV 2022.
>
> [2] Sigurdsson et al., "Hollywood in Homes: Crowdsourcing Data Collection for Activity Understanding," ECCV 2016.
>
> > **Q3**-The Methodology section lacks sufficient architectural details, A..., B..., C....
>
> A: We will clarify that **our method adopts a two-stage approach to extract holistic and partial features in the main paper.**
>
>
> B: We use ASFormer encoders and decoders according to ActDIFF to build the diffusion network, **where noisy input, timestep embeddings, and spatial, temporal, and frequency embeddings are first encoded using their own projection network (consisting of 2 linear layers) and then summed together.** Before the projection, timestep is first encoded with a sinusoidal positional embedding.
>
>
> C: **The BCA module enables information exchange between holistic and partial branches by using cross input-gate attention to modulate input information for each stream jointly, supporting richer temporal reasoning. HP-xLSTM leverages cross input-gate attention (i.e., BCA applied to input gates), enabling each branch to selectively incorporate temporal cues from the other based on feature-level gating.** This design enhances temporal segmentation and boosts per-frame accuracy, particularly in challenging multi-person scenarios with fine-grained actions.
>
>
> The attention weights and fused representations are computed as follows:
>
> $\alpha_t^{H \rightarrow P} = \text{softmax}\left(\frac{(W_Q X^H_t)(W_K X^P_t)^\top}{\sqrt{d}}\right)$
>
> $\tilde{X}^P_t = \alpha_t^{H \rightarrow P} \cdot (W_V X^H_t)$
>
> $\alpha_t^{P \rightarrow H} = \text{softmax}\left(\frac{(W_Q^* X^P_t)(W_K^* X^H_t)^\top}{\sqrt{d}}\right)$
>
> $\tilde{X}^H_t = \alpha_t^{P \rightarrow H} \cdot (W_V^* X^P_t)$
>
> Where $W_Q, W_K, W_V$ and  $W_Q^* , W_K^* , W_V^* $ are projections for queries, keys, and values for the two branches. $\tilde{X}^P_t$ and $\tilde{X}^H_t$ are partial and holistic inputs of HP-xLSTM. All the illustration in this response will be added into our manuscript.
>
> > **Q4**-Concerns of Innovation.
>
> We will enrich the discussion of dual-branch interaction and frequency-conditioned denoising. **Our novelty lies on the proposed holistic-partial aware xLSTM for temporal reasoning, where we propose bi-directional cross input-gate attention to enable holistic and partial aligned input control, and frequency condition, which is used to preserve temporal coherence and fine-grained details on motion dynamics by guiding the model to maintain consistency across both low and high-frequency components. Well-learned Fourier embeddings can better simulate temporal motion dynamics of specific frequencies.** The details of BCA information exchange are introduced in the response of Q3. Training two branches separately leads to better performance compared with the counterpart (two branches are fused together after HP-xLSTM) as in Table R4-A.
>
> **Table R4-A**
> |Method|Val ACC|Val EDIT|Val F1@10|Val F1@25|Val F1@50| Test ACC|Test EDIT|Test F1@10|Test F1@25|Test F1@50|
> |---------------------------|---------|----------|-----------|-----------|-----------|----------|-----------|------------|------------|------------|
> |Fused|52.12|**6.42**|87.10|84.51|81.49|56.50|**19.89**| 88.28| 87.69|86.64|
> |**Ours**|**54.41**|6.31|**89.14**|**86.56**| **83.35**|**59.63**|19.37|**90.11**|**90.33**| **89.26**|
>
> > **Q5**-Efficiency Issues.
>
> We agree that efficiency is crucial for practical use, and thus report trainable parameters, inference time, and GFLOPs alongside performance. **As shown in Table R4-A, HopaDIFF achieves state-of-the-art results on RHAS while being more efficient than methods like ASQuery and RefAtomNet, highlighting its superiority on RHAS tasks.**
>
> **Regarding the two-stage pipeline, we note that most existing methods (e.g., FACT, ActDIFF, LTContext, ASQuery) also adopt this approach due to the high computational cost of long video processing. We acknowledge this as an open challenge in the action segmentation field and will discuss it in the limitations and future work section.**
>
> **Table R4-B**
> |**Method**|**Trainable Param**|**Inference Time**|**GFLOPS**|**ACC (Test)**|**EDIT (Test)**|**F1@10 (Test)**|**F1@25 (Test)**|**F1@50 (Test)**|
> |----------------|---------------------|---------------------|-------------|-------------|-------------|-------------|-------------|-------------|
> |FACT|67.44M|1.98s|16.37G|27.89|0.44|59.05|58.37|57.13|
> |ActDIFF|7.7M|2.28s|7.14G|2.36|15.09|22.44|22.28|21.80|
> |LTContent|6.12M|2.70s|20.02G|52.52|0.37|49.35|47.24|42.55|
> |ASQuery|47.28M|1.24s|57.22G|20.84|0.08|33.86|32.86|30.20|
> |RefAtomNet|106M|3.29s|273.53G|38.73|0.13|41.16|39.17|35.44|
> |HopaDIFF|20M|2.81s|10.72G|**59.63**|**19.37**|**90.91**|**90.33**|**89.26**|
>
> > **Q6**-Performance and Fairness Issues:
>
> We recognize this as a limitation of existing diffusion action segmentation method. **As shown in Table 3 of the main paper, where a cross-movie partition is used, consistent with the setup in the module ablation, ActDIFF underperforms FACT. This suggests that the diffusion action segmentation model with only holistic features struggles to generalize across different movies.** Incorporating an additional partial branch, as in our approach, proves beneficial for enhancing the generalizability of diffusion-based action segmentation across diverse video styles. We will modify our paper accordingly.
>
>
> > **Q7**-Experimental Issues.
>
> A: For the cross-movie partition, **movies are randomly assigned to validation and test sets, preserving distribution shifts delivered by different movie styles.** In contrast, **under the random partition (Table 2 &4 in main paper), the performance gap between test and val sets is relatively small.**
>
>
> B: Since **our task focuses on fine-grained action annotations, recognizing precise action boundaries is inherently difficult,** leading to lower EDIT scores.
>
>
> C: As shown in Table R4-C, **removing the diffusion module causes a significant performance drop on EDIT score, highlighting its importance for action boundary reasoning.**
>
>
> D: **Module ablation results are provided in Table 5 in our main paper. The ASFormer encoder-decoder has 10.6M parameters, while HP-xLSTM adds around 9.4M.**
>
>
> E: Table R4-D presents the performance of each branch, demonstrating that **combining both holistic and partial branches yields the best overall performance.**
>
> **Table R4-C**
> |Method|Val ACC|Val EDIT|Val F1@10|Val F1@25|Val F1@50|Test ACC|Test EDIT|Test F1@10|Test F1@25|Test F1@50|
> |---------------------------|---------|----------|-----------|-----------|-----------|----------|-----------|------------|------------|------------|
> |w/o Diffusion|53.43|0.09|67.35|63.68|81.50|57.47| 58.56|0.10|56.17|53.72|48.92|
> |**Ours**|**54.41**|**6.31**|**89.14**|**86.56**| **83.35**|**59.63**|**19.37**|**90.11**|**90.33**| **89.26**|
>
> **Table R4-D**
> |Method|Val ACC|Val EDIT|Val F1@10|Val F1@25|Val F1@50|Test ACC|Test EDIT|Test F1@10|Test F1@25|Test F1@50|
> |---------------------------|---------|----------|-----------|-----------|-----------|----------|-----------|------------|------------|------------|
> |w/o partial branch|20.78|**6.34**|45.81|43.93|41.95|23.44|**19.93**| 50.60|50.14|49.42|
> |w/o holistic branch|44.48|6.28|79.40| 77.34|74.42|48.45|19.70|81.99|81.49|80.48|
> |**Ours**|**54.41**|6.31|**89.14**|**86.56**|**83.35** |**59.63**|19.37|**90.11**|**90.33**|**89.26**|
>
> > **Q8**-Errors and Inconsistencies.
>
> Thank you for pointing out the writing issues, we truly appreciate your effort. We will carefully correct the typos and inconsistent descriptions in the manuscript. Regarding the Fourier condition, yes, it should be “w/o DFT-cond,” as we use the DFT in our method.

---

> > ### Comment · Reviewer_dRDv · 2025-08-05
> >
> > The author’s response has addressed most of my concerns. Based on their reply and in consideration of the other reviewers’ comments, I have decided to raise my rating to 'accept.' Additionally, I would like the author to incorporate these improvements into the final manuscript.

---

> > > ### Author Response · Authors · 2025-08-05
> > > **Thank you**
> > >
> > > We sincerely appreciate your thoughtful feedback and valuable suggestions.  We will take great care to address all of your comments in the final version and your comments help us to improve the paper a lot.
> > > Thank you very much for the time and effort you invested in reviewing our work and for kindly raising your score, we are truly grateful. Once again, thank you for your support.

---

> ### Author Response · Authors · 2025-08-03
> **A Correction to Table R4-C in the Rebuttal**
>
> Dear Reviewer dRDv,
>
> First thank you very much for your comments again. During the rebuttal discussion with other reviewers we found there is an error made in Table R4-C of our responses to your questions.  We attach the corrected Table R4-C as follows, where the Test EDIT and Test F1@10 of w/o Diffusion variant should be switched in our previous response. We apologize for it in case there is any inconvenience.
>
> **Table R4-C**
> | Method        | Val ACC | Val EDIT | Val F1@10 | Val F1@25 | Val F1@50 | Test ACC | Test EDIT | Test F1@10 | Test F1@25 | Test F1@50 |
> |---------------|---------|----------|-----------|-----------|-----------|----------|-----------|------------|------------|------------|
> | w/o Diffusion | 53.43   | 0.09     | 67.35     | 63.68     | 81.50     | 57.47    | 0.10      | 58.56      | 56.17      | 53.72      |
> | **Ours**      | **54.41** | **6.31** | **89.14** | **86.56** | **83.35** | **59.63** | **19.37** | **90.11**  | **90.33**  | **89.26**  |
>
> Thank you very much for your review effort and please feel free to let us know if there is any other concern.

---

### Official Review · Reviewer_DuhU · 2025-06-23

**Clarity:** 3
**Significance:** 3
**Originality:** 3
**Rating:** 5
**Confidence:** 4

**Summary:**

The authors introduce Referring Human Action Segmentation (RHAS), a novel task for segmenting the actions of a specific individual in complex, multi-person videos guided by a textual reference. To facilitate research in this new area, the authors present the RHAS133 dataset. The primary technical contribution is HopaDIFF, a diffusion-based framework designed to address this task. The model integrates both holistic (full-frame) and partial (person-specific) information through a dual-branch architecture. Its key innovations include a novel cross-input gate attentional xLSTM to enhance long-range temporal reasoning and a Fourier-based conditioning mechanism to exert more precise control over the generation of action segments.

**Questions:**

The paper presents a novel and effective framework, but further clarification and analysis would benefit a few points.
- Could the authors provide qualitative results to offer a more complete picture of the model's performance?
- Could the authors replace HP-xLSTM with a more standard cross-attention mechanism (similar to that used in FACT [1]) to see if the advanced memory and gating features of xLSTM [2] are truly necessary for achieving state-of-the-art results?

Other details are reported in the weaknesses.

[1] Lu, Zijia, and Ehsan Elhamifar. "Fact: Frame-action cross-attention temporal modelling for efficient action segmentation." Proceedings of the IEEE/CVF Conference on Computer Vision and Pattern Recognition. 2024.

[2] Beck, Maximilian, et al. "xLSTM: Extended Long Short-Term Memory." Advances in Neural Information Processing Systems 37 (2025): 107547-107603.

**Ethical Concerns:**

["NO or VERY MINOR ethics concerns only"]

**Final Justification:**

I was already convinced of the potential of this paper, and the rebuttal and discussion phase further strengthened my confidence. The authors have thoroughly addressed all of my concerns. I continue to recommend this paper for acceptance.

**Limitations:**

Yes

**Quality:**

3

**Strengths And Weaknesses:**

## Paper Strengths
- **Well-Defined Motivation**: Most existing human action segmentation methods are tailored for single-person scenarios [1,2,3,4,5], where actions occur in a fixed sequence defined by a specific task. To address the limitations of these approaches in more complex settings, the authors propose a new task called Referring Human Action Segmentation (RHAS), which focuses on multi-person untrimmed videos guided by textual descriptions.
- **New Dataset Introduction**: The authors present a new dataset named RHAS133, which includes 133 untrimmed videos featuring a total of 542 individuals. Each person of interest is annotated with a corresponding textual reference, supporting the RHAS task.
- **Introduction of diverse baselines**: To highlight the difficulty of the RHAS task, the authors adapt and evaluate a range of existing state-of-the-art methods for temporal action segmentation, originally designed for single-person videos, within the RHAS setting. These adapted methods serve as a baseline for comparison and demonstrate the novel challenges of the task.
- **Introduction and Detailed Explanation of HopaDIFF**: The authors propose a novel framework named HopaDIFF (Holistic-Partial aware Fourier-conditioned Diffusion Framework). It includes two branches: a holistic branch that captures the global context from the entire video, and a partial branch that uses GroundingDINO to locate the referred individual, generating a cropped video for fine-grained representation via a Vision-Language Model (VLM). To support efficient reasoning over long videos and integrate holistic and partial information, the authors introduce the HP-xLSTM module, a modified version of the mLSTM [6] module that incorporates a bidirectional cross-attention (BCA) mechanism at the input gate. Uniquely, this framework integrates Fourier conditioning, offering finer control over the generative process by operating in the frequency domain.
- **Effectiveness of HopaDIFF**: As shown in Section 5 of the paper, HopaDIFF significantly outperforms baseline methods, demonstrating strong performance gains. The authors conduct an ablation study, analyzing the impact of different VLM backbones and showing that while backbone quality matters, HopaDIFF consistently outperforms competitors. Additional ablations highlight the individual contributions of each component within the framework.

## Paper Weaknesses
- **Absence of Qualitative Results**: The paper's evaluation is entirely quantitative. While the reported metrics are strong, the paper would be significantly improved by including qualitative results. Visual comparisons are essential for building intuition about the model's performance on this complex task. It would be beneficial for the authors to include:
    - Side-by-side visualizations of the segmentation output from HopaDIFF compared to key baselines on selected video segments.
    - A dedicated analysis of failure cases to provide insight into the model's current limitations.
- **Limited Scope of Ablation Studies**: The provided ablation study effectively demonstrates the contribution of each component within the HopaDIFF framework. However, a more compelling analysis would involve testing the modularity of these components by integrating them into the baseline architectures. This would provide stronger, more isolated evidence of their value. Specific suggestions include:
    - *Integrating the Partial Branch into FACT [1]*: Since the FACT [1] baseline also uses a dual-branch architecture, its action branch could be replaced with the proposed GroundingDINO-based partial reasoning branch. This would directly test the effectiveness of this component outside of the full HopaDIFF model.
    - *Augmenting ActDiff with Frequency Conditioning*: The authors note that the ActDiff [2] baseline operates on temporal and spatial features. A valuable experiment would be to augment ActDiff [2] with the proposed DFT-based frequency conditions to determine if this mechanism provides a general performance benefit to diffusion-based action segmentation models.
    - *Comparing HP-xLSTM to Standard Attention*: The ablation study shows a performance drop when removing the HP-xLSTM module. However, to better justify its complexity, it would be insightful to replace HP-xLSTM with a more standard cross-attention mechanism (similar to that used in FACT [1]) to see if the advanced memory and gating features of xLSTM [6] are truly necessary for achieving state-of-the-art results.

$\underline{\text{Minor Revision}}$:
In Figure 2, there is a typo: Patial -> Partial.

[1] Lu, Zijia, and Ehsan Elhamifar. "Fact: Frame-action cross-attention temporal modeling for efficient action segmentation." Proceedings of the IEEE/CVF Conference on Computer Vision and Pattern Recognition. 2024.

[2] Liu, Daochang, et al. "Diffusion action segmentation." Proceedings of the IEEE/CVF international conference on computer vision. 2023.

[3] Bahrami, Emad, Gianpiero Francesca, and Juergen Gall. "How much temporal long-term context is needed for action segmentation?." Proceedings of the IEEE/CVF International Conference on Computer Vision. 2023.

[4] Gan, Ziliang, et al. "ASQuery: A Query-based Model for Action Segmentation." 2024 IEEE International Conference on Multimedia and Expo (ICME). IEEE, 2024.

[5] Peng, Kunyu, et al. "Referring atomic video action recognition." European Conference on Computer Vision. Cham: Springer Nature Switzerland, 2024.

[6] Beck, Maximilian, et al. "xLSTM: Extended Long Short-Term Memory." Advances in Neural Information Processing Systems 37 (2025): 107547-107603.

---

> ### Author Rebuttal · Authors · 2025-07-30
>
> Dear Reviewer DuhU,
>
> Thank you for recognizing the contribution of our work, your comments and the review effort. We provide point-to-point responses to your concerns as follows,
>
> > **W1**-Absence of Qualitative Results: The paper's evaluation is entirely quantitative. While the reported metrics are strong, the paper would be significantly improved by including qualitative results. Visual comparisons are essential for building intuition about the model's performance on this complex task.
> >  It would be beneficial for the authors to include:
> > 1. Side-by-side visualizations of the segmentation output from HopaDIFF compared to key baselines on selected video segments.
> > 2. A dedicated analysis of failure cases to provide insight into the model's current limitations.
> >
> > **Q1**-Could the authors provide qualitative results to offer a more complete picture of the model's performance?
>
> Thank you very much for your valuable suggestion. We fully agree that including more qualitative results and a detailed analysis of failure cases can provide deeper insight into the strengths and limitations of our proposed method and the newly introduced task.
>
> During the rebuttal phase, we prepared a set of qualitative results and failure case analyses to address this point. However, as stated in the official email from the conference, sharing images during the rebuttal is strictly prohibited. Therefore, we were unable to include these visual results in our response. But we still deliver the observations as follows.
>
> For the qualitative results, we conducted a comparative analysis among our proposed method (HopaDIFF), ActDIFF, and FACT. We observed that HopaDIFF consistently produces action segmentation outputs that more closely align with the ground truth labels. **Notably, our method demonstrates superior performance in handling periodic action transitions along temporal axis. This improvement can be attributed to the high granularity of the condition offered by the Fourier embeddings, especially on periodic temporal patterns, and the enhanced temporal reasoning capabilities of our proposed holistic-partial aware cross input-gate xLSTM architecture towards different visual focus.**
>
> In terms of failure case analysis, we identified a performance drop in classes with limited training samples. **In such long-tail scenarios, frames corresponding to rare classes are often misclassified as more frequent classes. We acknowledge this limitation and consider tackling the long-tail distribution challenge in referring human action segmentation as a promising and important direction for future research.**
>
> We will incorporate these qualitative comparisons and failure analyses in the final version of our manuscript to provide a more comprehensive evaluation of our approach.
>
> > **W2**-Limited Scope of Ablation Studies:  The provided ablation study effectively demonstrates the contribution of each component within the HopaDIFF framework. However, a more compelling analysis would involve testing the modularity of these components by integrating them into the baseline architectures. This would provide stronger, more isolated evidence of their value. Specific suggestions include:
> > 1. Integrating the Partial Branch into FACT [1]: Since the FACT [1] baseline also uses a dual-branch architecture, its action branch could be replaced with the proposed GroundingDINO-based partial reasoning branch. This would directly test the effectiveness of this component outside of the full HopaDIFF model.
> > 2. Augmenting ActDiff with Frequency Conditioning: The authors note that the ActDiff [2] baseline operates on temporal and spatial features. A valuable experiment would be to augment ActDiff [2] with the proposed DFT-based frequency conditions to determine if this mechanism provides a general performance benefit to diffusion-based action segmentation models.
> > 3. Comparing HP-xLSTM to Standard Attention: The ablation study shows a performance drop when removing the HP-xLSTM module. However, to better justify its complexity, it would be insightful to replace HP-xLSTM with a more standard cross-attention mechanism (similar to that used in FACT [1]) to see if the advanced memory and gating features of xLSTM [6] are truly necessary for achieving state-of-the-art results.
>
> > **Q2-** Could the authors replace HP-xLSTM with a more standard cross-attention mechanism (similar to that used in FACT [1]) to see if the advanced memory and gating features of xLSTM [2] are truly necessary for achieving state-of-the-art results?
>
>
> Thank you very much for your comment. We agree with you that the mentioned ablations are interesting and could enhance our experimental validation of the proposed approach. We thereby provide more ablations in Table R3-A.
>
> **Table R3-A: More Ablation Results**
>
> | Method                    | Val Acc | Val EDIT | Val F1@10 | Val F1@25 | Val F1@50 | Test Acc | Test EDIT | Test F1@10 | Test F1@25 | Test F1@50 |
> |---------------------------|---------|----------|-----------|-----------|-----------|----------|-----------|------------|------------|------------|
> | FACT [37]        | 26.01   | 0.34 | 60.16 | 57.79 | 55.33 | 27.89   | 0.44 | 59.05 | 58.37 | 57.13 |
> | FACT w/ partial branch         | 18.35   | 0.32     | 62.57     | 61.10     | 61.10     | 18.34    | 0.43      | 63.72      | 63.35      | 62.40      |
> | ActDiff [31]     | 4.46    | 5.96 | 38.86 | 38.36 | 37.51 | 3.26    | 15.09| 22.44 | 22.28 | 21.80 |
> | ActDIFF w/ DFT-cond         | 10.46   | 6.29     | 26.61     | 24.80     | 23.40     | 11.92    | 19.54     | 30.01      | 29.64      | 28.92      |
> | Cross-Attention Fusion           | 51.74   | **6.41**     | 86.91     | 84.47     | 81.50     | 57.02    | **19.87**     | 88.97      | 88.41      | 87.36      |
> | **Ours**         | **54.41** | 6.31 | **89.14** | **86.56** | **83.35** | **59.63** | 19.37 | **90.11** | **90.33** | **89.26** |
>
>
>
> ### 1. FACT ablation with partial branch. ###
> When comparing FACT and FACT w/ partial branch, formulated as suggested, we observe that substituting the action branch with the partial branch leads to improvements in F1@{10,25,50}. However, this substitution also causes a decline in the ACC metric. This is because the holistic branch captures broader interaction-related actions, such as those involving communication with other individuals or background context. Replacing it with the partial branch results in the loss of important contextual information. On the other hand, the partial branch shifts the learning focus to the target individual, which enhances recognition of personal movement-related actions. Movement-related actions, such as sitting, typically span longer temporal durations. **As a result, replacing the action branch with the partial branch helps the model better capture these temporally extended patterns by concentrating on the motion of the target individual.
> This targeted focus improves the model’s ability to accurately localize long-duration actions over time, resulting in higher temporal intersection over union and consequently better performance on F1@{10,25,50}, which accounts for temporal overlap in evaluation. However, this shift in focus comes at the cost of overlooking broader contextual cues, which are important to the actions with communication with background context, resulting in a trade-off in performance, as reflected in the ACC metric.**
> These dynamics explain the performance variations shown in Table R3-A.
>
> ### 2. ActDIFF ablation with frequency condition. ###
> We present an ablation study combining ActDIFF with the frequency-domain condition in Table R3-A. **Compared to the original ActDIFF, this variant shows improvements across most evaluation metrics, with a notable gain in the EDIT score. This suggests that the frequency condition helps the ActDIFF model better capture action boundaries during segmentation. However, we also observe a drop in F1 scores on the validation set. Given that the validation and test sets are drawn from different movies, this decline indicates that, without the support of the HP-xLSTM module, the frequency condition alone has limited generalization capability across varying movie contexts in the referring human action segmentation task.**
>
>
>
> ### 3. Compare HP-xLSTM with standard cross attention. ###
>
> In our ablation study, we compare the variant using Cross-Attention Fusion with our HopaDIFF. The results show that cross-attention fusion is indeed effective, achieving 51.74% Acc, 6.41% EDIT, and F1 scores of 86.91% (F1@10), 84.47% (F1@25), and 81.50% (F1@50) on the validation set. On the test set, this variant reaches 57.02% Acc, 19.87% EDIT, and F1 scores of 88.97%, 88.41%, and 87.36% at the respective thresholds.
>
> **While this variant performs comparably to HopaDIFF with HP-xLSTM in terms of the EDIT score, it falls short on other key metrics, particularly Acc and F1 across all thresholds. This performance gap highlights the effectiveness of our proposed HP-xLSTM module, which offers a more structured and fine-grained mechanism for long-range temporal modeling and bidirectional information exchange between the holistic and partial branches.** Unlike cross-attention fusion, which applies attention globally between branches, HP-xLSTM introduces cross-input gated attention that enables each branch to selectively incorporate temporal controls from the other on their individual features from gating perspective. This leads to better temporal segmentation and improved per-frame accuracy, particularly in complex, multi-person scenarios with fine-grained actions.
>
> > Minor revision
>
> The mentioned typo will be corrected in our manuscript.
>
>
> We sincerely thank the reviewer once again for the insightful ablation suggestions, which help to enrich the ablation design and analyses of our paper. All corresponding experiments and analyses will be incorporated into the revised manuscript.

---

> ### Comment · Reviewer_DuhU · 2025-08-03
>
> Many thanks to the authors for their detailed response. The answer to the question regarding the limited scope of ablation studies effectively underscores the importance of each component within the proposed HopaDIFF framework.
>
> However, after reading the other reviewers' feedback, a point of confusion remains regarding the response to Reviewer rUnz's question:
>
> > Q3-Regarding the evaluation protocol, if there are multiple ground truth action labels for the same person at the same timestamp (e.g., sitting and watching for person 1, Fig. 1), how do the authors determine whether the action segmentation is correct? Should the method output all ground truth labels, or is it sufficient to output only one of them?
>
> The authors state that they compare the top-k predicted labels with the ground-truth annotations, where k is the number of true labels for that specific frame. While this is a valid approach, it diverges from the standard multi-label classification paradigm where the number of true labels is unknown at test time. To provide clearer context for future work, the authors can discuss this methodological choice in the limitations section.
>
> More critically, there appears to be a discrepancy between the described evaluation logic and the implementation. The authors state:
> > If the predicted top-k set exactly matches the ground-truth labels for that frame, we treat the prediction for that frame as correct.
>
> However, the corresponding line in the code the authors provide in the supplementary materials, `if np.sum(np.sort(ind_l) == np.sort(ind_p))>=1:`, does not appear to check for an exact match. Instead, this condition is satisfied if there is at least one common label between the predicted set and the ground-truth set. This implementation is substantially more lenient than an "exact match" metric.
>
> Could the authors please clarify this point?

---

> > ### Author Response · Authors · 2025-08-03
> > **Response to Reviewer DuhU**
> >
> > Dear Reviewer DuhU,
> >
> > Thank you very much for your response!
> >
> > We appreciate our conducted experiments have addressed your concerns regarding the ablation study.
> >
> > >The authors state that they compare the top-k predicted labels with the ground-truth annotations, where k is the number of true labels for that specific frame. While this is a valid approach, it diverges from the standard multi-label classification paradigm where the number of true labels is unknown at test time. To provide clearer context for future work, the authors can discuss this methodological choice in the limitations section.
> >
> > Yes, we agree with your observation and will include a corresponding discussion in the limitations section.
> > Unlike standard multi-label classification paradigms, where each sample typically requires a single multi-label prediction, our task involves generating multi-label predictions for each frame to evaluate human action segmentation using metrics such as ACC, EDIT, and F1@{10,25,50}. In this context, setting a fixed value of k, as is commonly done in typical multi-label classification tasks, is not practically applicable, especially given that our dataset contains 137 classes.
> >
> > Thank you very much for your insightful suggestion.
> >
> >
> > > However, the corresponding line in the code the authors provide in the supplementary materials, if np.sum(np.sort(ind_l) == np.sort(ind_p))>=1:, does not appear to check for an exact match. Instead, this condition is satisfied if there is at least one common label between the predicted set and the ground-truth set. This implementation is substantially more lenient than an "exact match" metric.
> >
> > Thank you for pointing that out!
> >
> > We also became aware of this issue during the rebuttal phase, the utils.py file previously uploaded to the anonymous GitHub repository is an outdated version of the code, and we unfortunately overlooked updating it. However, according to the policy, we were not allowed to make any changes now.
> >
> > We would like to clarify that all of our experiments were conducted using the criterion if np.sum(np.sort(ind_l) == np.sort(ind_p)) == len(ind_l) on our local server, which enforces exact matching. The authors have carefully checked the codes in local. We sincerely apologize for any confusion this may have caused and appreciate your effort in reviewing the code. We will update our github repository as soon as the conference gives us allowance.

---

> > > ### Comment · Reviewer_DuhU · 2025-08-03
> > >
> > > Thank you very much for your thoughtful and detailed response. With these clarifications, I will maintain my **Accept** rating and continue to support the paper for acceptance.

---

> > > > ### Author Response · Authors · 2025-08-03
> > > > **Thank you**
> > > >
> > > > Thank you very much for your insightful feedback and helpful suggestions.
> > > >
> > > > We truly appreciate the time and effort you dedicated to reviewing our work and helping us improve it.
> > > >
> > > > Your comments have provided valuable guidance, and we will take them all into careful consideration as we prepare the final paper. Thank you very much.

---

### Official Review · Reviewer_rUnz · 2025-06-26

**Clarity:** 2
**Significance:** 3
**Originality:** 3
**Rating:** 4
**Confidence:** 4

**Summary:**

The paper introduces the first work for addressing temporal action segmentation in multi-person scenarios.
The authors constructed a new dataset specifically for this new task.
They further propose a diffusion-based solution that integrates both global (holistic) and local (partial) image features, with the latter capturing the specific target subject being described.
The solution also leverages frequency-domain information to enhance diffusion-based action segmentation.
Experiments on the proposed dataset show that the proposed solution outperforms baseline methods.
The ablation study further verifies the benefits of incorporating partial image features and leveraging frequency-domain information.

**Questions:**

1. When training and testing existing methods (i.e., FACT, ActDiff, ASQuery, LTContent, RefAtomNet), what are the inputs for these methods?
    - Do they take the original images containing multiple people as input, or do they use image crops featuring only the person being described?
    - Or I would say, what are the performances of existing methods if we first crop the target person based on textural description and then feed these image crops to these existing methods?
2. As a follow-up to the first question, what if considering an ablated setup of the proposed solution, where this ablated setup does not leverage holistic features? Examining its performance could illustrate the benefits of capturing interactions among people through holistic features, which could further justify the importance of the proposed task.
3. Regarding the evaluation protocol, if there are multiple ground truth action labels for the same person at the same timestamp (e.g., sitting and watching for person 1, Fig. 1), how do the authors determine whether the action segmentation is correct? Should the method output all ground truth labels, or is it sufficient to output only one of them?
4. Why does leveraging frequency-domain information benefit action segmentation? It would be beneficial if the authors could explain the motivation for this design choice.
5. For the ablated setting _w/o HP-xLSTM_ and _w/o BCAF_, compared to the full solution, what are their differences in the detailed framework design?

**Ethical Concerns:**

["NO or VERY MINOR ethics concerns only"]

**Final Justification:**

The rebuttal has addressed most of my concerns raised in my original comments. I am glad to raise my score towards a positive rate.

**Limitations:**

Yes, the authors have adequately addressed the limitations and potential negative societal impact in their submission.

**Paper Formatting Concerns:**

There is no formatting issue.

**Quality:**

2

**Strengths And Weaknesses:**

**Strengths**:
- The paper addresses the important task of multi-person action segmentation, which has not been discussed in previous works.
- The paper contributes a new dataset designed for this important task.
- The proposed solution leverages image features of description-conditioned image crops and captures frequency domain information, which are proven to be effective.

**Weakness**:
- The paper lacks necessary detailed explanations regarding the implementation of existing methods and ablated settings. Regarding this issue, I have further explained in the question session.
- I consider that the authors should discuss a naive solution that first crops the target subject based on the input textural reference and then solves by applying existing single-person action segmentation frameworks on these image crops.

Therefore, I vote for "borderline reject" for the current version. I hope the authors could address my concerns listed in the weakness and questions sessions.

---

> ### Author Rebuttal · Authors · 2025-07-30
>
> Dear Reviewer rUnz,
>
> Thank you for taking the time to review our work and your valuable suggestions. Below, we provide our detailed, point-by-point responses to your comments.
>
> > **W1**-Implementation of the existing approaches.
>
> Thank you for your comment. **We clarify that all the baselines in our main paper use the video with whole-image video as input.** We provide further point-to-point answer to your each question in this response.
>
> > **W2**-using cropped images of target person as input.
>
> Thank you for your suggestion. **The ablation results in Table R2-A show that using video with only cropped person regions leads to a consistent drop in ACC compared with using whole-image video, with partial improvements in F1 scores, showing that embeddings learned on the whole video differs from the cropped video, however the leveraged single person baselines still deliver unsatisfactory performances.** Since the actions involves not only person physical movement (e.g., sitting) but also involves actions communicate with other persons and background objects (e.g., talking to). Using only cropped video will cause the performance drop on actions of the latter case.
>
>
> > **Q1**-When training and testing existing methods (i.e., FACT, ActDiff, ASQuery, LTContent, RefAtomNet), what are the inputs for these methods?
>
> Thank you for your comment. **All the baseline approaches take the original images (with multi persons and w/o cropping) together with textual reference to a desired person as input.** We will clarify these details in our manuscript.
>
> > * 1. Do they take the original images containing multiple people as input, or do they use image crops featuring only the person being described?
>
> Thank you for the comment. **All the baselines take the original images containing multiple people as input.** We will clarify this in our manuscript.
>
> > * 2. Or I would say, what are the performances of existing methods if we first crop the target person based on textural description and then feed these image crops to these existing methods?
>
> Thank you for the valuable comment. We agree that incorporating these ablation studies strengthens our experimental analysis. Accordingly, we provide Table R2-A to present the results. To ensure a fully automated pipeline and avoid manual intervention, the bounding boxes used for cropping are obtained from GroundingDINO.
>
> When comparing the performance of existing methods that use only cropped person regions with Table 3 in our main paper, **we observe a consistent decline in the ACC metric across all baselines, along with partial improvements in F1 scores.** These findings highlight the complementary roles of both holistic and partial branches in referring human action segmentation. **Specifically, the holistic branch incorporates background and contextual information, which aids in recognizing actions involving multi-person communication and human-object interaction. In contrast, the partial branch focuses on the referred individual, enhancing recognition and segmentation of personal movement-related actions.**
>
> **Table R2-A: Ablation experiments with cropped video as input, when we use BLIP-2 feature extractor and cross-movie partition**
> | **Method** | **ACC (Val)** | **EDIT (Val)** | **F1@10 (Val)** | **F1@25 (Val)** | **F1@50 (Val)** | **ACC (Test)** | **EDIT (Test)** | **F1@10 (Test)** | **F1@25 (Test)** | **F1@50 (Test)** |
> |-------------------------|---------------|----------------|------------------|------------------|------------------|----------------|------------------|-------------------|-------------------|-------------------|
> | FACT| 16.73| 0.39| 66.07| 65.73| 64.69| 15.82| 0.30| 57.30| 55.75| 53.72|
> | ActDIFF| 3.84| 3.51| 51.38| 50.53| 47.87| 2.04           | 7.67| 44.13| 43.70| 42.42|
> | LTContent| 45.36| 0.08|60.51|57.00|50.87| 50.34          | 0.08| 49.58| 47.14| 42.43|
> | ASQuery| 32.42| 0.11| 51.18| 48.31| 43.32| 36.51| 0.12   | 44.46| 42.52| 39.09|
> | RefAtomNet| 27.73| 0.13| 46.90| 43.52|38.82| 30.45          | 0.15|38.01| 36.30| 32.72|
> | Ours w/ only partial branch                | 44.48     | 6.28       | 79.40        | 77.34        | 74.42        | 48.45      | **19.70**        | 81.99         | 81.49         | 80.48         |
> | **Ours w/ holistic and partial branches**| **54.41** | **6.31** | **89.14** | **86.56** | **83.35** | **59.63** | 19.37 | **90.11** | **90.33** | **89.26** |
>
>
> > **Q2-** A follow-up to the first question, what if considering an ablated setup of the proposed solution, where this ablated setup does not leverage holistic features? Examining its performance could illustrate the benefits of capturing interactions among people through holistic features, which could further justify the importance of the proposed task.
>
> Thank you very much for this comment which helps us to explore more on HopaDIFF. We conducted the suggested ablation experiment and compare it with our approach in Table R2-B. **We observe that removing the holistic branch brings large performance degradation on the evaluation metrics, i.e., ACC, F1@10, F1@25, and F1@50, illustrating that the holistic information provided by the whole original images is important for the model to understand the human actions in the complex scenario, especially considering the actions which need the target person communicate with the environment and other persons**.
>
>
> **Table R2-B: Ablation experiments for HopaDIFF**
> | **Method**|**ACC (Val)**|**EDIT (Val)**|**F1@10 (Val)** | **F1@25 (Val)**|**F1@50 (Val)** |**ACC (Test)** | **EDIT (Test)**|**F1@10 (Test)**|**F1@25 (Test)** | **F1@50 (Test)**|
> |-------------------------|---------------|----------------|------------------|------------------|------------------|----------------|------------------|-------------------|-------------------|-------------------|
> |**Ours (w/o holistic branch)**|44.48     |6.28|79.40|77.34|74.42|48.45|**19.70**|81.99| 81.49|80.48|
> | **Ours**|**54.41**|**6.31**|**89.14**|**86.56**| **83.35** |**59.63**|19.37|**90.11**|**90.33**| **89.26** |
>
>
>
> > **Q3**-Regarding the evaluation protocol, if there are multiple ground truth action labels for the same person at the same timestamp (e.g., sitting and watching for person 1, Fig. 1), how do the authors determine whether the action segmentation is correct?
> > Should the method output all ground truth labels, or is it sufficient to output only one of them?
>
> Thank you for your thoughtful question.
> In this work, we adopt standard action segmentation metrics, including per-frame accuracy (ACC), EDIT distance, and F1 scores at temporal IoU thresholds of 10%, 25%, and 50%. Since fine-grained actions often involve multiple co-occurring labels (e.g., "sitting" and "watching"), **we adapt these metrics to a multi-label setting**.
>
> To account for this, **we evaluate the model’s predictions on a per-frame basis by comparing the top-k predicted labels with the ground-truth annotations, where k is the number of ground-truth labels provided for that frame. If the predicted top-k set exactly matches the ground-truth labels for that frame, we treat the prediction for that frame as correct**. This true positive judgement is used in every aforementioned metric.
>
> > **Q4**-Why does leveraging frequency-domain information benefit action segmentation? It would be beneficial if the authors could explain the motivation for this design choice.
>
> Thank you for your question. We are happy to elaborate on the motivation behind leveraging frequency-domain information as condition for diffusion based human action segmentation.
>
> In our task of referring human action segmentation, accurately identifying the temporal transitions of fine-grained actions is critical. Traditional diffusion-based models rely only on temporal and spatial feature conditions to guide the generation process. However, such representations may not capture the full spectrum of temporal dependencies, especially subtle or periodic patterns that are important for fine-grained action understanding.
>
> To address this, we incorporate frequency-domain information by applying the DFT along the temporal axis of the learned spatial and temporal embeddings. **Fourier condition allows the diffusion action segmentation model to be conditioned on global temporal characteristics, such as periodic motion patterns or transitions between actions, which are not always apparent in the raw time domain.** By conditioning the diffusion process with these frequency-based cues, the model gains finer control over the generation process, improving both the temporal precision and stability of the action segmentation outputs.
>
> Our ablation study (Table 5 in the paper) confirms the benefit of this design choice: removing frequency-domain conditioning leads to a notable drop in F1 scores across all thresholds. This validates the frequency information's role in enhancing segmentation quality, particularly in complex, multi-person scenarios.
>
> > **Q5**-For the ablated setting w/o HP-xLSTM and w/o BCAF, compared to the full solution, what are their differences in the detailed framework design?
>
> Thank you for the question. In the w/o HP-xLSTM ablation, we remove the entire HP-xLSTM module, meaning that there is no more xLSTM based temporal reasoning in the framework. We directly use the extracted features from BLIP-2 as input to the ASFormer encoder and decoder to formulate the diffusion training and inference process. This limits the model’s ability to integrate global and person-specific cues, reducing temporal alignment and segmentation quality.
>
> In the w/o BCAF ablation, we keep the HP-xLSTM structure for temporal modeling, but we disable the cross-branch attention gating at the input stage. The experimental results in Table 5 in our main paper show that both ablations result in decreased performance, especially in F1 scores, highlighting the importance of both the HP-xLSTM architecture and the BCAF mechanism in enabling strong long-range, target-aware temporal modeling.

---

> > ### Comment · Reviewer_rUnz · 2025-08-03
> >
> > I appreciate the feedback from the authors, and consider that the feedback has addressed most of my concerns. I now lean towards accepting this submission.

---

> > > ### Author Response · Authors · 2025-08-03
> > > **Thank You**
> > >
> > > We sincerely appreciate your thoughtful feedback and constructive suggestions. Your comments have been very helpful for improving our paper, and we will carefully incorporate all of your comments into the final version.
> > >
> > > Thank you so much for your effort and for raising your score. We truly appreciate it. Thank you once again.

---

### Official Review · Reviewer_YL35 · 2025-07-03

**Clarity:** 3
**Significance:** 4
**Originality:** 3
**Rating:** 5
**Confidence:** 4

**Summary:**

This paper introduces **Referring Human Action Segmentation (RHAS)**, a novel task that focuses on segmenting the actions of specific individuals in multi-person untrimmed videos based on textual references. To support this task, the authors present **RHAS133**, a new dataset comprising 133 movies and 542 annotated individuals. They also propose **HopaDIFF**, a diffusion-based framework that integrates holistic and partial visual reasoning with Fourier-based conditioning to enhance action segmentation performance.

**Questions:**

Please refer to **Strengths And Weaknesses**.

**Ethical Concerns:**

["NO or VERY MINOR ethics concerns only"]

**Final Justification:**

For my part, I believe the authors have adequately addressed my main concerns. Having reviewed the other reviewers' comments, which are all positive, I am inclined to maintain my recommendation as Accept.

**Limitations:**

Please refer to **Strengths And Weaknesses**.

**Quality:**

4

**Strengths And Weaknesses:**

### **Strengths**

1. **Novel and Well-Motivated Task**: The RHAS task addresses a real and underexplored limitation in existing action segmentation methods—namely, the over-reliance on single-person scenarios with fixed protocols. The integration of referring expressions into multi-person action segmentation is novel and practically relevant, making the problem both original and impactful.

2. **Valuable Dataset Contribution**: The RHAS133 dataset is, to the best of our knowledge, the first specifically designed for referring human action segmentation. It provides rich annotations including 137 fine-grained action classes, natural language referring expressions, and multi-label annotations, making it a valuable and reusable benchmark for the community.

3. **Reasonable Technical Approach**: The proposed dual-branch (holistic-partial) architecture is well-motivated, aiming to jointly capture global contextual cues and target-specific fine-grained motion patterns. The use of HP-xLSTM with a cross-input gating mechanism provides a sensible way to facilitate information exchange between branches.

4. **Thorough Experimental Evaluation**: The experimental section is comprehensive, including a wide range of adapted baselines from both action segmentation and referring action recognition literature. In addition, the evaluation under both random partition and cross-movie splits offers useful insights into model generalization across settings.

---

### **Weaknesses**

1. **Limited Technical Novelty**: The overall architecture mainly combines existing components—such as diffusion models, xLSTM, and DFT-based conditioning—rather than proposing fundamentally new algorithmic techniques. As such, the contribution is more integrative than innovative.

2. **Technical and Methodological Concerns**: The model involves several components (dual branches, HP-xLSTM, BCA, and DFT conditioning), but the paper lacks sufficient justification or ablation for each design choice. Furthermore, there is no discussion or analysis of computational complexity, inference latency, or comparison with more lightweight alternatives, which raises concerns about scalability and deployment.

---

> ### Author Rebuttal · Authors · 2025-07-30
>
> Dear Reviewer YL35,
>
> Thank you for the review effort and recognizing the merit of our work. We have provided a point-to-point response as follows.
>
> > **W1**-Limited Technical Novelty: The overall architecture mainly combines existing components, such as diffusion models, xLSTM, and DFT-based conditioning, rather than proposing fundamentally new algorithmic techniques. As such, the contribution is more integrative than innovative.
>
> Regarding the novelty of our architecture, **we would like to clarify that our HopaDIFF is the first model to propose a cross input-gate holistic-partial aware xLSTM strategy for long-term temporal reasoning that explicitly incorporates both global video context and partial features specific to the referred individual. In addition, we introduce a novel Fourier-based condition mechanism that enhances the condition granularity by considering periodic temporal patterns from global temporal perspective in diffusion-based action segmentation.**
> To the best of our knowledge, both of these components, cross input-gate attention within xLSTM for long term temporal reasoning between holistic and partial branches, and frequency-domain conditions are introduced for the first time for diffusion based human action segmentation.
>
> In our experiments, we observed that existing human action segmentation methods struggle with the RHAS task, especially due to its fine-grained annotation and the complexity of multi-person, in-the-wild video scenarios. Although the diffusion-based method ActDiff achieves relatively strong performance in terms of temporal boundary modelling (e.g., high evaluation performance in terms of  EDIT score), its results vary significantly across different evaluation settings. This highlights a key challenge: the need for more expressive and fine-grained condition to support stable and accurate action segmentation prediction generation. Motivated by this, we designed the HP-xLSTM module to strengthen long-term temporal modeling, enabling bidirectional information flow between holistic and partial branches through cross input-gate attention.
> Notably, our use of cross-attention at the input gate level represents a novel contribution to xLSTM design.
> HP-xLSTM incorporates cross input-gate attention, allowing each branch to selectively integrate temporal cues from the other based on feature-level gating. This mechanism enhances temporal segmentation and improves per-frame accuracy, especially in complex multi-person scenarios involving fine-grained actions.
> Complementing this, we incorporate frequency domain information, via discrete Fourier transform to improve temporal coherence and control. Frequency conditioning, as we propose it here, allows the model to account for both low- and high-frequency patterns, which is critical for precise action segmentation in challenging real-world settings.
>
> Beyond the model itself, we would also like to emphasize our major contributions to the field. This work introduces a new task, i.e., referring to human action segmentation in multi-person scenarios, and contributes the first dataset for this purpose, RHAS133.
> The dataset was carefully curated and annotated over the course of eight months and includes fine-grained frame-level labels guided by textual references.
> We believe this benchmark fills an important gap in the literature and offers a solid foundation for future research in language-guided video understanding in unconstrained environments.
>
> > **W2**-Technical and Methodological Concerns: The model involves several components (dual branches, HP-xLSTM, BCA, and DFT conditioning), but the paper lacks sufficient justification or ablation for each design choice. Furthermore, there is no discussion or analysis of computational complexity, inference latency, or comparison with more lightweight alternatives, which raises concerns about scalability and deployment.
>
> We appreciate the reviewer’s concern on the technical details and provide the response as follows.
>
> ### Ablation of individual component:
> Each module, dual branches, HP-xLSTM, BCA, and DFT-based conditioning, was carefully designed to address specific challenges in referring human action segmentation, such as the need for fine-grained, temporally consistent predictions for a target individual in complex multi-person scenes and the capability to deal with periodic temporal patterns. **The effectiveness of these components is demonstrated through detailed ablation studies in Table R1-A (corresponding to Table 5 in our manuscript), where removing any of the modules leads to significant performance drops.** The ablation study evaluates five variants of our model: (1) w/o all, a baseline diffusion-based action segmentation model without any of our proposed components; (2) w/o HCMGB, which removes the GroundingDINO-based partial reasoning branch; (3) w/o HP-xLSTM, which omits the holistic-partial aware xLSTM temporal aggregation module; (4) w/o BCAF, which excludes bidirectional cross-attention between the input gates of the two branches; and (5) w/o DFT-cond, which removes frequency-domain conditioning in both branches. For example, eliminating the partial branch causes F1@50 to fall from 89.26% to 49.42%, highlighting the critical role of localized, target-aware features.
> Similarly, excluding HP-xLSTM or the frequency-based conditioning results in consistent degradation in segmentation quality, showing that each component contributes materially to the final performance.
>
> **Table R1-A: Ablations on RHAS133 using BLIP-2 and cross-movie partition**
>
> | **Method**         | **ACC (Val)** | **EDIT (Val)** | **F1@10 (Val)** | **F1@25 (Val)** | **F1@50 (Val)** | **ACC (Test)** | **EDIT (Test)** | **F1@10 (Test)** | **F1@25 (Test)** | **F1@50 (Test)** |
> |--------------------|---------------|----------------|------------------|------------------|------------------|----------------|------------------|-------------------|-------------------|-------------------|
> | w/o all            | 4.46          | 5.96           | 38.86            | 38.36            | 37.51            | 2.36           | 15.09            | 22.44             | 22.28             | 21.80             |
> | w/o HCMGB          | 20.78         | 6.34           | 45.81            | 43.93            | 41.95            | 23.44          | **19.93**            | 50.60             | 50.14             | 49.42             |
> | w/o HP-XLSTM       | 47.75         | 6.30           | 82.81            | 80.61            | 77.63            | 52.01          | 19.68            | 85.06             | 84.54             | 83.50             |
> | w/o BCAF           | 44.24         | **6.35**       | 79.76            | 77.38            | 74.50            | 49.56          | 19.87            | 82.98             | 82.45             | 81.38             |
> | w/o DFT-cond       | 50.84         | 6.33           | 85.88            | 83.47            | 80.52            | 55.49          | 19.92            | 87.95             | 87.42             | 86.38             |
> | **Ours**           | **54.41**     | 6.31           | **89.14**        | **86.56**        | **83.35**        | **59.63**      | 19.37        | **90.91**         | **90.33**         | **89.26**         |
>
>
> ### Efficiency of the proposed approach:
>
> We thank the reviewer for raising this valuable point. We agree that including efficiency measurements helps clarify the practicality of the proposed model.
> To this end, we report the number of trainable parameters, inference time, and GFLOPS for each method alongside their performance.
> **As shown in Table R1-B, HopaDIFF achieves superior performance on the RHAS task while requiring fewer trainable parameters and lower computational cost compared to methods like ASQuery and RefAtomNet. These results highlight the effectiveness of HopaDIFF in balancing accuracy and efficiency for referring human action segmentation.**
>
> **Table R1-B: Efficiency measurement when we leverages BLIP-2 feature extractor on RHAS133, cross movie partition**
> | **Method**     | **Trainable Param** | **Inference Time** | **GFLOPS**  | **ACC (Test)** | **EDIT (Test)** | **F1@10 (Test)** | **F1@25 (Test)** | **F1@50 (Test)** |
> |----------------|---------------------|---------------------|-------------|-------------|-------------|-------------|-------------|-------------|
> | FACT           | 10.29M              | 1.98s            | 16.37G      | 27.89   | 0.44     | 59.05     | 58.37     | 57.13     |
> | ACTDIFF        | 7.7M                | 2.28s            | 7.14G       | 2.36    | 15.09    | 22.44     | 22.28     | 21.80     |
> | LTContent      | 6.12M               | 2.70s            | 20.02G      |52.52   | 0.37     | 49.35     | 47.24     | 42.55     |
> | ASQuery        | 47.28M              | 1.24s            | 57.22G      |20.84   | 0.08     | 33.86     | 32.86     | 30.20     |
> | RefAtomNet     | 106M                | 3.29s            | 273.53G     | 38.73   | 0.13     | 41.16     | 39.17     | 35.44     |
> | HopaDIFF       | 20M             | 2.81s        | 10.72G  |**59.63** | **19.37** | **90.91** | **90.33** | **89.26** |

---

> ### Comment · Reviewer_YL35 · 2025-08-05
> **Response to the rebuttal**
>
> For my part, I believe the authors have adequately addressed my main concerns. Having reviewed the other reviewers' comments, which are all positive, I am inclined to maintain my recommendation as Accept.

---

> > ### Author Response · Authors · 2025-08-05
> > **Thank you**
> >
> > We truly appreciate your insightful feedback and valuable suggestions. Your comments have been instrumental in guiding the improvement of our paper, and we will thoughtfully incorporate them into the final version. Thank you very much again for your review effort and time.

---

### Note · Authors · 2025-08-12

Dear AC and Reviewers,

Thank you for taking the time to review our paper. We would like to provide a brief summary of the rebuttal process for your reference.

We sincerely appreciate the reviewers’ thorough evaluation and constructive comments. All reviewers actively engaged in the discussion and offered valuable perspectives. We are especially encouraged by their acknowledgment of the strengths and contributions of our work.

**Task:** All reviewers recognized the novelty and significance of our proposed referring human action segmentation task, noting that our work is the first to address human action segmentation with textual reference in unconstrained, multi-person scenarios, with comments such as "both original and impactful" (YL35), "important task" (rUnz), "Well-Defined Motivation" (DuhU), and "of high significance to the field" (dRDv).

**Dataset and Benchmark:** In this work, we proposed the first dataset for the RHAS task, i.e., RHAS133. The new dataset is also acknowledged by all the reviewers, with comments such as "Valuable Dataset Contribution" (YL35). The benchmark effort is recognized by the reviewers with comments "Thorough Experimental Evaluation" (YL35), and "Introduction of diverse baselines" (DuhU).


**Methodology:** The proposed method, HopaDIFF, were praised by all reviewers, with comments such as "Reasonable Technical Approach" (YL35), "proven to be effective" (rUnz), "strong performance" (DuhU), and "The technical methodology is sound" (dRDv).


During the rebuttal and author–reviewer discussion phases, we are pleased to report that most of the reviewers’ concerns have been effectively addressed through our detailed responses and additional experiments. We have conducted the requested ablation studies and provided further clarifications to highlight the significance of our proposed approach, along with additional model details, such as trainable parameters, inference time, and GFLOPs, to ensure a fair and transparent efficiency comparison.

We sincerely thank you again for your time and thoughtful consideration of our submission. We hope that our comprehensive rebuttal and the improvements made will contribute to a positive final evaluation. We will incorporate all constructive feedback into the final version, which are detailed in our responses to each reviewer to further strengthen the quality, clarity, and impact of the work.

Sincerely,

The authors

---

### Decision · Program_Chairs · 2025-09-17

**Decision:**

Accept (spotlight)

**Comment:**

This paper proposes a new dataset, RHAS133, for textual reference-guided human action segmentation in multi-person settings. Based on the dataset, a new method is designed and evaluated. The method is based on a diffusion model for action segmentation, incorporating multiple modifications such as cross-input attention and Fourier conditions.

Reviewers commented positively regarding the novel task proposed, the new dataset contributed, the reasonable method design, and the comprehensive experiments. Reviewers commented negatively mainly on the limited technical novelty, combining existing components, missing ablation studies of complicated components, and the absence of qualitative results. The weaknesses have been well addressed after the rebuttal.

The main reason for acceptance is the new task and the new dataset, advancing the field of action segmentation. Action segmentation has been saturated for a relatively long time, and the classic formulation does not keep up with vision-language foundation models. This paper, with its new task and dataset, will provide new opportunities for the community.

The dataset, evaluations, and codes should be fully released as a benchmark to facilitate future research.